



# Compositions and mixing states of aerosol particles by aircraft observations in the Arctic springtime, 2018

Kouji Adachi[1], Naga Oshima[1], Sho Ohata[2,3,4], Atsushi Yoshida[2], Nobuhiro Moteki[2], and Makoto Koike[2]

[1] Department of Atmosphere, Ocean, and Earth System Modeling Research, Meteorological Research Institute, Tsukuba, Japan
[2] Department of Earth and Planetary Science, Graduate School of Science, The University of Tokyo, Tokyo, Japan
[3] Institute for Space–Earth Environmental Research, Nagoya University, Nagoya, Japan
[4] Institute for Advanced Research, Nagoya University, Nagoya, Japan

*Correspondence to*: Kouji Adachi (adachik@mri-jma.go.jp)

**Abstract.** Aerosol particles were collected at various altitudes in the Arctic during the Polar Airborne Measurements and Arctic Regional Climate Model Simulation Project (PAMARCMiP 2018) conducted in the early spring of 2018. The composition, size, number fraction, and mixing state of individual aerosol particles were analyzed using transmission electron microscopy (TEM), and their sources and transport were evaluated by numerical model simulations. We found that sulfate, sea-salt, mineral-dust, K-bearing,

and carbonaceous particles were the major aerosol constituents and were internally mixed. The number fraction of mineral-dust and sea-salt particles decreased with increasing altitude. The K-bearing particles increased within a biomass burning (BB) plume at altitudes > 3900 m, which originated from Siberia. Chlorine in sea-salt particles was replaced with sulfate at high altitudes. These results suggest that the sources, transport, and aging of Arctic aerosols largely vary depending on the altitude and airmass history.

We also provide the occurrences of solid-particle inclusions (soot, fly-ash, and Fe-aggregate particles), some of which are light-absorbing and potential ice-nucleating particles. Our TEM measurements revealed, for the first time, the detailed mixing state of individual particles at various altitudes in the Arctic. This information facilitates the accurate evaluation of the aerosol influences on Arctic haze, radiation balance, cloud formation, and snow/ice albedo when deposited.





## 1 Introduction

The Arctic is sensitive to climate change. The surface-air temperature in this region is rising more rapidly than the global average, resulting in decreases in seasonal sea ice coverage and thickness (Stroeve et al., 2012). Aerosol particles are an important factor influencing the Arctic climate (Abbatt et al., 2019). They

scatter and absorb solar radiation and affect cloud formation. Light-absorbing aerosol particles decrease the snow albedo when deposited on snow surfaces (Hansen and Nazarenko et al., 2004). Hence, understanding their chemical and physical properties, such as size, abundance, composition, and mixing state, is required to accurately evaluate the influence of aerosol particles on the climate in this region.

The sources of the aerosol particles within the Arctic region include anthropogenic activities, the ocean,

glaciers, and ground surfaces. In general, the local anthropogenic sources within the Arctic are limited because of the limited human activities. However, air pollution over the Arctic named Arctic haze due to long-range transport (LRT) aerosols from mid-low latitudes has been observed in winter and spring (Stone et al., 2014; Shaw et al., 1995; Law et al., 2014; Law and Stohl, 2007). These LRT aerosol particles originate from, for example, anthropogenic emissions from urban cities in East Asia, North America, and

Europe, biomass burning (BB) from Siberia and North America, and desert areas (Abbatt et al., 2019; Lathem et al., 2013; Hecobian et al., 2011).

In the Arctic, the near-surface air temperature is low at lower altitudes, resulting in a temperature inversion, thereby forming a polar dome with little vertical mixing and removal of aerosols (Law et al., 2014: Arnold et al., 2016; Shaw et al., 1995). LRT particles travel along several pathways from their

source regions and reach beyond the polar dome (Law et al., 2014; Schulz et al., 2019). As a result, LRT particles are difficult to monitor via ground measurements. Thus, aircraft observations have been conducted over the Arctic during field observation campaigns such as the Arctic Research of the Composition of the Troposphere from Aircraft and Satellites (ARCTAS), Aerosol, Radiation, and Cloud Processes affecting Arctic Climate (ARCPAC), Arctic Study of Tropospheric Aerosol, Clouds and

Radiation (ASTAR), and Network on Climate and Aerosols: Addressing Key Uncertainties in Remote Canadian Environments (NETCARE) (Law et al., 2014; Arnold et al., 2016; Brock, et al., 2011; Yamanouchi et al., 2005; McNaughton et al., 2011; Willis et al., 2019; Wendisch et al., 2019; Jacob et al., 2010).

LRT particles are subject to mixing processes with other aerosol particles or gas condensation processes

during transport. The composition and mixing state of individual particles largely differ from their original ones. These differences influence their optical and hygroscopic properties, thus altering their climate effects. For example, soot particles coated with sulfate or organic matter exhibit an enhanced light absorption by the focusing of incoming light on the soot core, and the coatings may change them into efficient cloud condensation nuclei (CCN) (Bond et al., 2013). Information on the soot-mixing state is



also essential in climate models to accurately evaluate their atmospheric lifetime (i.e., their removal or transport efficiency) and, in turn, their direct radiative effects (Matsui and Moteki, 2020; Moteki et al., 2019).

These mixing processes occur on the surfaces of individual particles with size ranges of hundreds of nanometers. Single-particle mass spectrometry (PALMS; Murphy et al., 2006; Brock et al., 2011) and
microscopy measurements have been performed to examine the composition and mixing state of individual aerosol particles in the Arctic. Brock et al. (2011) investigated the individual aerosol composition in the Arctic during the ARCPAC aircraft campaign using PALMS. In addition, various microscopy measurements of Arctic aerosol particles have been conducted using samples collected during ground observations (e.g., Anderson et al., 1992; Geng et al., 2010; Chi et al., 2015). However, electron
microscopy measurements at various altitudes using an airplane have been absent during recent campaigns except that performed by Hara et al. (2003) during the ASTAR 2000 campaign. The concentration of black carbon (BC) or sulfate in the Arctic atmosphere has changed over the past decades (Gong et al., 2010), and sea ice melting is accelerating (Stroeve et al., 2012). Thus, knowledge of the individual particles in the current Arctic atmosphere is essential to better understand the influence of
aerosol particles on the Arctic climate.

We used transmission electron microscopy (TEM) for single-particle analysis of aerosol particles at various altitudes in the Arctic. TEM is an off-line technique and requires aerosol particle samples collected with TEM grids, thus resulting in a low time resolution and loss of volatile materials. On the other hand, TEM directly measures the shape, composition, mixing state, and inclusions within host
particles. TEM also directly identifies solid particles consisting of soot, metals, or dust, which potentially become ice-nucleating particles (INPs; Krämer et al., 2017).

The current study was conducted as part of the Polar Airborne Measurements and Arctic Regional Climate Model Simulation Project 2018 (PAMARCMiP 2018) in the early spring of 2018. The overall goal of this campaign is to better understand and quantify the interaction between atmospheric aerosols, surface
optical properties, and clouds in the central Arctic. In this campaign, Yoshida et al. (2020) measured the size, abundance, and mixing state of BC and Fe-oxide particles using a single-particle soot photometer (SP2). Hartmann et al. (2020) also reported that particles from marine sources exhibit enhanced INP activities and characterized the aerosol particles within high-INP activity samples using TEM. The current study focused on TEM analysis and measured the composition and mixing state of aerosol particles at
various altitudes. The goal of this study is to investigate the detailed individual-particle composition and mixing state to better understand aerosol climate influences, aging, and LRT in the Arctic.



## 2 Methods

### 2.1 Campaign and TEM sampling

The PAMARCMiP 2018 campaign was conducted from 23 March to 4 April 2018 using the research

aircraft Polar 5 (Wesche et al., 2016) to observe aerosols, sea ice, and clouds. The aircraft was based at the Villum Research Station (81°36 N, 16°40 W) at Station Nord in Greenland. Fourteen flights were executed during the campaign above the Arctic Ocean and the Fram Strait at various altitudes up to ~5200 m (Fig. 1).

TEM samples were collected using an aerosol sampler (AS-16W, Arios Inc., Tokyo, Japan) with two

impactor stages. The airflow was 1.0 L/min with 0.5- and 1.0-mm nozzle sizes for the fine and coarse modes, respectively. The sampled particle sizes are ~0.1-0.7 μm in aerodynamic diameter (50% cutoff size) at the fine stage and >0.7 μm at the coarse stage. The sampler mounted 16 TEM grids under both the fine and coarse modes during each research flight. TEM 200-mesh Cu grids with a formvar carbon substrate (U1007, EM-Japan, Tokyo, Japan) were applied for sample collection. The sampling time was

19 min during regular flights and 29 min during long flights (April 3 and 4) with a 1-min pause time between each sampling. In total, we collected 139 TEM samples during the campaign. The samples were sealed after each flight and stored under dry conditions at room temperature.

### 2.2 TEM measurements

We recorded ~30 TEM images per each grid from all TEM samples under the fine mode and selected 26

TEM samples from eight research flights to cover the various altitudes and periods of interest for detailed composition and mixing state analysis (Table S1). A transmission electron microscope (JEM-1400, JEOL, Tokyo, Japan) equipped with an energy-dispersive X-ray spectrometer (EDS; X-Max 80 mm, Oxford Instruments, Tokyo, Japan) was used for both TEM- and scanning TEM (STEM)-mode analyses. A semiautomatic particle measurement system using the STEM-EDS measured the particle

size and composition at an acceleration voltage of 120 kV and a 20-sec acquisition time. More than 200 particles around the center parts under the jet nozzle were analyzed for each sample. As the particle number concentration in the sampled air was low, the collected particles were dispersed and hardly overlapped on the substrates. The particle size was measured based on the area-equivalent diameter (AED) determined from the STEM images. The AED may be larger than the aerodynamic diameter for

particles with a flat shape or low density. The AED was obtained for each particle by assuming that it only consists of its representative composition with the recognition that most particles are internally mixed and contain several components. Elemental mapping images of representative particles were also acquired using STEM-EDS. Details of the STEM-EDS analysis have been provided elsewhere (Adachi et al., 2019, 2020).



The STEM-EDS analysis normalizes the element composition to 100% among the selected elements (C, N, O, Na, Mg, Al, Si, P, S, Cl, K, Ca, V, Cr, Mn, Fe, Zn, and Pb). The detection limits of each element are determined based on one sigma of the peak intensity and calculated with EDS software (INCA Energy version 5.02, Oxford Instruments, Tokyo, Japan). This study considers only nonvolatile materials such as mineral-dust, sea-salt, K-bearing, sulfate, and non- or low-volatile carbonaceous

particles (e.g., soot and tarballs). Volatile materials such as volatile organic compounds, nitrate, and water are lost after sampling and in the TEM chamber and are thus not considered. We categorize the analyzed particles into six particle types based on their composition, i.e., mineral-dust particles (both Al and Fe > 0.2 wt.%), sea-salt particles (Na > 1 wt.%), K-bearing particles (K > 2 wt.%), sulfate particles (S > 2 wt.%), carbonaceous particles (C + O > 90 wt.%), and other particles. The workflow of particle

classification is shown in Fig. S1. This classification may underestimate the particles in the lower categories in the flow chart, such as sulfate and carbonaceous particles, which partially coat or are attached to other particles. The classification is the same as that used by Adachi et al. (2020) except for P-bearing particles, a proxy of primary biological aerosol (PBA) particles. The P-bearing particles are not included in the current study because of the lack of PBA particles.

In addition to the particle-type category based on the composition, we identified the inclusions embedded within the particles. Soot, fly-ash, and Fe-oxide aggregate (hereafter Fe-aggregate) particles were identified as inclusions based on their shape and composition. Soot particles consist of aggregates of ~40-nm nanospheres possessing structures of concentrically wrapped, graphene-like layers. They have been defined as nanosphere-soot when analyzed using TEM (Buseck et al., 2014). We adopt the

definition of nanosphere-soot for our soot particles and employ the term soot for simplicity. We also use the term BC instead of soot in the model simulations and for the results from those instruments that measure BC by assuming that soot and BC are qualitatively the same material. An individual Fe-aggregate particle usually consists of more than two spherical iron-oxide (magnetite) particles (Moteki et al., 2017; Yoshida et al., 2018; Ohata et al., 2018; Kurisu et al.; 2019; Li et al., 2017). Fe-aggregate

particles mainly originate from anthropogenic sources (Moteki et al., 2017). Fly-ash particles exhibit spherical shapes and occur as either single particles or aggregates. They mainly consist of Si, Al, and Fe and originate from anthropogenic combustion sources. Nonaggregated Fe particles were categorized as fly-ash particles.

## 2.3 Model simulations

We applied the Meteorological Research Institute Earth System Model version 2 (MRI-ESM2; Yukimoto et al., 2019; Oshima et al., 2020) to estimate the possible source regions and transport pathways of the mineral-dust, sea-salt, and soot particles sampled by the aircraft. The model configurations are basically the same as those in Adachi et al. (2020) except for the simulation period



and the classification of the BC sources. In the current study, the simulation period was extended to
2018 (from January 2008 to December 2018), and the BC concentrations were further divided into those
originating from anthropogenic and BB sources. We used an atmospheric general circulation model
(AGCM) with land processes (MRI-AGCM3.5) and the Model of Aerosol Species in the Global
Atmosphere mark-2 (MASINGAR mk-2) as atmospheric and aerosol component models, respectively.
The model treats non-sea-salt sulfate, BC, organic carbon, sea salt, mineral dust, and aerosol precursor
gases. The model uses a horizontal resolution with an approximately 120-km grid (TL159) and 80
vertical layers from the surface to the model top at 0.01 hPa in a hybrid sigma-pressure coordinate
system. The horizontal wind fields were nudged toward the 6-hourly Japanese 55-year Reanalysis data
(Kobayashi et al. 2015) to reproduce realistic meteorological fields in the simulations. We used the
daily BB emissions from the global fire assimilation system dataset of Kaiser et al. (2012) and the
monthly anthropogenic emissions dataset of Lamarque et al. (2010). In addition to the baseline
simulation, we performed a model sensitivity simulation without BC emissions from BB sources (i.e.,
anthropogenic BC). The BC concentration from BB sources (hereafter referred to as BB BC) was
estimated by subtracting the anthropogenic BC concentration (the sensitivity simulation) from the total
BC concentration (the baseline simulation). The emissions of mineral dust and sea salt were calculated
based on the meteorological conditions in the simulations (Tanaka and Chiba, 2005; Yumimoto et al.,
2017).

## 3 Results and discussion

### 3.1 Number fraction and size of the major aerosol types

The aerosol particles in our samples mainly consist of sulfate, sea-salt, mineral-dust, K-bearing, and
carbonaceous particles. They are mixed at the single-particle scale and sometimes contain inclusions.
We measured the particle-size distribution and particle number fraction of each aerosol type (Figs. 2
and 3). We classified the samples based on the sampling altitude below 1000 m and above 1000 m. The
former was mainly influenced by local emissions, and the latter included LRT aerosol particles.

The median diameter of all particles is 470 nm ($\sigma = 410$) in the AED (Fig. 2). The sulfate, K-bearing,
and carbonaceous particles exhibit similar size distributions (the median AEDs are 470, 450, and 400
nm, respectively), whereas the carbonaceous particles demonstrate a sharper peak than do the others.
The mineral-dust and sea-salt particles attain larger and broader size distributions than do the others
with median diameters of 680 and 740 nm, respectively.

Sulfate particles are the most abundant in all samples except the sample collected on 31 March, 14:40,
which predominantly contains sea-salt particles. The sulfate number fraction ranges from 29% to 86%
(59% on average) (Table S1; Fig. 3). The second most dominant particle types are sea-salt and





carbonaceous particles for the samples collected at < 1000 m and > 1000 m, respectively. Potassium-bearing particles are relatively abundant (accounting for > 10% of the number fraction) in three samples collected at > 3900 m on 2, 3, and 4 April (Figs. 4 and 5). Based on the TEM observations and model

simulations, we found that the above K-bearing particle-rich samples were collected from air masses influenced by BB, and we denoted them as BB samples (please refer to section 3.2 for the transport of the BB plume).

The size-dependent number fractions (Fig. 3) indicate that sulfate particles dominate the < 1 µm particles in the samples collected below 1000 m. In the samples collected above 1000 m, sulfate

particles dominate all bin sizes except the largest size bin (> 2 µm), which only contains 12 particles and attains a low statistical significance (Fig. 3 (b)). The fraction of mineral-dust particles increases with increasing particle size. The number fractions of the sea-salt and carbonaceous particles increase with increasing and decreasing particle size, respectively. In contrast to the negligible contributions of the K-bearing particles in the samples collected below 1000 m, their number fraction in the samples

collected above 1000 m is ~10% in all size bins except the largest one.

The number fraction of the aerosol types varies depending on the sampling altitude. The mineral-dust and sea-salt number fractions decrease with increasing sampling altitude (Fig. 4). The K-bearing particle fraction is high in the BB samples collected above 3900 m. The number fractions of the sulfate and carbonaceous particles did not show a clear altitude dependency in this study, although other

studies have demonstrated that their absolute number concentrations depend on the altitude (e.g., Brock et al., 2011).

### 3.2 K-bearing particles and biomass burning plume

K-bearing particles are the second dominant type (11-37% in number fraction) in the BB samples (Fig. 5 and Table S1). The K-bearing particles mainly occur as potassium sulfate mixed with organic matter,

soot, or both (Fig. 6). Other than in the K-bearing particles, K is detected as a minor component in approximately half of all particles and 76% of the soot-bearing particles.

BB smoke contains a large amount of K in addition to other aerosols, such as organic matter and soot particles (Reid et al., 2005), and K can be used as a tracer element of BB. The occurrences of the K-bearing particles in the BB samples (Fig. 6) are similar to those from BB in other areas (e.g., Li et al.,

225    2003).

We used the modeled BB BC to identify the transport pathways of the BB samples (Fig. 7). The BB BC emissions in the model simulations show that the BB source is southeast Siberia (Fig. S2), where forest BB frequently occurs (Brock et al., 2011; Warneke et al., 2009; Schulz et a, 2019). During the campaign, BB started on approximately 20 March in the area and persisted for several months. On 28

March, the emitted BB BC plume was driven toward the northeast by a low-pressure system.



Thereafter, the BB plume was lifted and subsequently transported toward the North Pole until 1 April (Fig. S2). On April 2, 3, and 4, a part of the BB plume approached the Fram Strait at ~600 hPa (~4000 m), where we conducted the sampling. Although the model simulations estimated the BB contributions only for the BC concentrations, we interpret that the K-bearing particles and other BB emissions were

also transported along with BC as seen in the TEM results. In terms of the BB samples, the sampling points were > 4600 km away from the BB sources, and the samples had aged for a week or more.

### 3.3 Sea-salt particles

Sea-salt particles are globally abundant, especially over oceans, and considerably influence the climate as CCN and via sunlight scattering (Lewi and Schwartz, 2004). They are formed from seawater film

droplets over the open ocean and local leads in the Arctic. When the sea freezes, sea-salt particles form on frost flowers over sea ice (Hara et al., 2017; Xu et al., 2016) and from blowing snow (Huang and Jaeglé, 2017). Our samples could originate from these sources as our research flights flew over both open water (near Svalbard islands) and sea ice (near Greenland) (Fig. 1). Original sea-salt particles mainly consist of sodium chloride as well as other inorganic salts (e.g., Mg, Ca, and K as chlorides or

sulfates). In the atmosphere, the composition of sea-salt particles is altered through the reactions with acidic gases, thus forming sodium sulfate or nitrate (Adachi and Buseck, 2015; Gard et al., 1998; Yoshizue et al., 2019).

In our samples, the sea-salt particles are complex mixtures of, for example, sodium chloride, sodium sulfate, magnesium sulfate, Mg-C-O, calcium sulfate, and others (Fig. 8 and Fig. S3). The sea-salt

occurrences are similar to those found at a ground site in Svalbard (Chi et al., 2015). Although nitrate can react with sea-salt particles, N was rarely detected in our sea-salt particles, possibly because the nitrate fraction relative to that of sulfate is limited in spring (Brock et al., 2011; Fenger et al., 2013). It is also possible that the measured particle sizes are too small for nitrate to retain as a particle phase and that nitrates are lost from the TEM samples after sampling because of their high volatility.

Some Mg occurs around NaCl cores (Fig. 8a-b). These mixtures may form either in the atmosphere or on the substrate when liquid particles change into solid phases after sampling. Such Mg occurs with C and O as an amorphous phase, suggesting that they constitute organic matter. Similar organic Mg-C coatings on sea-salt particles have been observed in the Arctic (Chi et al., 2015). As Mg salts exhibit a high hygroscopicity, it is possible that they are liquid in the atmosphere and absorb organic matter either

from anthropogenic or marine sources (Shaw et al., 2010). Some sea-salt particles demonstrate homogeneous distributions of Mg, S, and Na (Fig. 8c and Fig. S3e-f). These particles commonly attach soot particles, and Cl is replaced with sulfate, suggesting that they are well aged in the atmosphere.

Most Cl in the sea-salt particles was replaced with sulfate in the samples collected above 1000 m (Fig. S4). Such Cl loss in NaCl at high altitudes in the Arctic has also been observed by Hara et al. (2002).





Mg and Ca are correlated with Na in all samples, although their weight percent is lower in the samples collected above 1000 m than those collected below 1000 m (Fig. 9). This result suggests that most Mg, Ca, and Na originates from sea-salt particles and that their weight percent within the individual particles decreases with increasing sampling altitudes because of the condensation and coagulation of sulfate or other material on the sea-salt particles.

In the model simulations, the sea-salt concentrations near the sea surface are high (Fig. S5). On 30 and 31 March, the sea-salt particle concentrations are higher than those on the other sampling days based on the TEM analysis (Table S1), and the model simulation indicates that the sea-salt particles are transported from the north.

### 3.4 Mineral-dust particles

Certain types of mineral-dust particles (e.g., feldspar)  act as INPs (Kanji et al., 2017) and have a climatological impact on the Arctic (Fan, 2013). In our samples, the number fraction of the mineral-dust particles increases with increasing particle size (Fig. 3) and decreasing altitude (Fig. 4). They mainly consist of Si, Al, or both and include Na, Mg, K, Ca, and Fe as minor components or small grains (e.g., Fe) (Fig. 10). They exhibit irregular shapes with crystalline structures. The mineral-dust particles are

commonly mixed with sea-salt and sulfate particles, both of which change the mineral-dust particles from hydrophobic to hydrophilic. Such mixtures have also been observed in low-latitude regions (e.g., the Amazon basin) in other studies (Adachi et al., 2020), suggesting that mixtures of mineral dust and hygroscopic particles (e.g., sea salt and sulfate) occur globally. The Na and Cl weight percent within the mineral-dust particles increases in the samples collected below 1000 m (Fig. S6), indicating that

mineral-dust and sea-salt particle mixing occurs near the surface. These results suggest that the mineral-dust particles originate from local ground surfaces. However, although mineral-dust particles may have local sources in summer, such as glacial outwash planes (Tobo et al., 2019) and bare ground surfaces, the ground surfaces in Greenland or the Svalbard islands were mostly, if not completely, covered by snow and ice during the sampling period (March and April). Our model simulation does not include

mineral-dust emissions in the Arctic during the sampling period because of the snow coverage and yields only a small amount of LRT mineral dust above ~700 hPa in the sampling area (Fig. S7). Long-term ground observations of the aerosol composition at station Nord have indicated small contributions from soil particles in the early spring, although they exhibit peaks in summer (Nguyen et al., 2013; Heidam et al., 1999). McNaughton et al. (2011) measured the dust mass concentration over the Western

Arctic in spring during the ARCTAS/ARCPAC 2008 campaigns and demonstrated that the mineral-dust concentration was high at > 4 km in altitude, where an Asian dust plume possibly influenced. They also showed that mineral-dust particles decrease toward the surface, although the sea-salt mass vertical distributions were consistent with our results. In contrast to these previous studies, our observation





suggests that mineral-dust particles could be emitted from local ground surfaces, but the source of the

mineral-dust particles in our samples cannot be confirmed. As the number fractions of the mineral-dust

particles are nonnegligible (4% on average), further observations are necessary to identify their sources.

### 3.5 Sulfate particles

Sulfate particles are one of the most dominant aerosol species in the Arctic (Brock et al., 2011; Hara et

al., 2003; Matsui et al., 2011). This study also showed that sulfate particles were the most dominant

species (~60%) in their number fraction (Fig. 3). The sulfate particle number fractions are not well

correlated with the altitude ($R^2$=0.16), suggesting that the detected sulfate particles and their precursors

originate from both marine surface sources (Willis et al., 2017) and LRT at high altitudes. Sulfate

commonly mixes with other species, and S is detected in almost all sampled particles. Sulfate particles

generally exhibit various chemical forms, including ammonium sulfate, sodium sulfate, potassium

sulfate, and calcium sulfate (Figs. 6 and 10; Fig. S3). The sulfate particles commonly have satellite

structures around their rims in the TEM images (Fig. 11), suggesting that sulfuric acid and ammonium

bisulfate contribute to the sulfate particles (Kojima et al., 2004). This result is consistent with other

measurements in the Arctic that show contributions of acidic sulfate (Brock et al., 2011; Hara et al.,

2003; Fisher et al., 2011).

### 3.6 Carbonaceous particles

Carbonaceous particles primarily consist of C and O and include secondary and primary organic

aerosols, soot particles, and tarballs. Secondary organic aerosols coat or embed other species and are

mostly classified into the categories of their host species. Volatile organic compounds are likely lost

before the analysis, and thus, their number fraction may be underestimated over the mass fraction

measured using online instruments. Tarballs or tarball-like particles are spherical organic particles

originating from BB (Pósfai et al., 2004) (Fig. 12) and are a major aerosol type originating from BB as

brown carbon (Chakrabarty et al., 2010; Sedlacek et al., 2018). As a result, they potentially influence

the climate, although their detailed occurrences, including their removal processes, remain unknown. In

the BB samples, we encountered a small number of tarball particles (<1 % in number fractions). These

tarballs mainly consist of C and O and include some N and K (Fig. 12), and their composition is similar

to that of particles from young BB smoke plumes (e.g., Pósfai et al., 2004; Adachi and Buseck, 2011).

Moroni et al. (2017, 2020) also found tarballs with K-bearing particles using scanning electron

microscopy in ground observations on the Svalbard Islands in the summer of 2015. In contrast to the

smooth spherical shapes of fresh tarballs (e.g., Li et al., 2003), the surfaces of our tarballs are not

smooth and contain sulfates (Fig. 12), suggesting that they reacted with other species and had aged

during LRT.





Soot particles were also commonly found in our samples. However, only 13% of the carbonaceous particles consisted of soot with thin or no coatings, i.e., external mixtures, and they were mainly embedded within or attached to other particles (internal mixtures) and were classified as a part of their
host species.

### 3.7 Soot, fly-ash, and Fe-aggregate inclusions

Inclusions are particles embedded within or attached to host particles. Inclusions are identified based on both their composition and shape determined from the TEM images after the removal of beam-sensitive materials (e.g., sulfate) by exposure to an electron beam during the STEM-EDS analysis (Fig. 13 and
Fig. S8). When they overlap with non-beam-sensitive materials (e.g., mineral dust and sea salt), it is difficult to detect inclusions in the TEM images. Thus, the actual number fraction of the inclusions may be larger than that reported in this study. On the other hand, this technique has the advantage of identifying small inclusions (e.g., < 50 nm), which are difficult to detect using other methods. Although we observed several metal particles as inclusions (e.g., Zn and Pb), we focused on the occurrences of
soot, fly-ash, and Fe-aggregate particles in this study.

### 3.7.1 Soot inclusions

Soot particles absorb solar radiation and exert a positive radiative forcing (Bond et al., 2013). When they are coated by or embedded within other species (e.g., sulfate and organic matter), their light absorption is enhanced by the focusing of light on the soot core (Bond et al., 2006; Oshima et al., 2009).
Studies involving Arctic BC observations have shown a 54% increase in BC light absorption due to its coating (Zanatta et al., 2018) or an approximately 20% increase in the direct radiative effect due to the different BC mixing state assumptions (Kodros et al., 2018). In addition, although soot particles are initially hydrophobic, such coatings commonly make them hydrophilic. The degrees of the soot mixing states with hygroscopic materials influence their atmospheric lifetime in models, resulting in uncertainty
in the simulated atmospheric BC concentrations in the Arctic (Eckhardt et al., 2015). Thus, the mixing states of soot particles are of interest regarding their climate influences (Fierce et al., 2020).

In our samples, soot particles were found in ~17% of all measured particles (Fig. S9 and Table S1). Most soot particles were internally mixed with other particles (Fig. 13 and Fig. S8). The number fraction of the externally mixed soot particles, which are those without any apparent coatings, was only
~1% of all soot particles. The relatively low number fractions of the externally mixed soot particles are similar to the background conditions observed by Hara et al. (2003) and indicate that most LRT soot particles were internally mixed in the Arctic.

Soot particles were observed within all aerosol types, and the sampling altitude was not clearly correlated with the soot particle fraction ($R^2$=0.13). On the other hand, more soot particles were found





in large host particles than in small ones, i.e., the size distributions of the soot-bearing particles (host particles) were larger than those of all particles (Fig. S10). This result is consistent with those in other areas (Adachi and Buseck, 2008; Adachi et al., 2014). A possible explanation is that the coagulation process of soot particles with large host particles occurs more efficiently than that with small particles.

BB is one of the major sources of BC in the Arctic atmosphere and snow (Spackman et al., 2010; Hegg

et al., 2010). In our BB samples, ~93% of the soot-bearing particles contained K, showing substantial BC contributions from BB. In comparison, the ratio was 74% for the non-BB samples. However, there was no apparent enhancement of the soot particle number fraction in the BB samples (Table S1). This inconsistency may be explained by the fact that we measured the number fraction and that the other species, such as the K-bearing salts, sulfates, and organic matters, also increased within the BB plume,

resulting in relatively lower soot number fractions in the BB samples.

### 3.7.2 Fly-ash and Fe-aggregate inclusions

The fly-ash particles and primary particles of Fe aggregates commonly exhibited spherical shapes, which indicated that they formed through rapid cooling after melting or evaporation at high temperatures during emission. The shape and composition of the Fe aggregates are consistent with those

observed near sources in East Asia (Moteki et al., 2017). Moteki et al. showed that Fe-aggregate particles consist of magnetite, which absorbs light and imposes warming effects on the climate. Yoshida et al. (2020) also demonstrated light-absorbing Fe-oxide particles using SP2 during this campaign. Thus, we suggest that our Fe-aggregate particles also exhibit light-absorbing properties.

The number fractions of the particles containing fly-ash or Fe-aggregate particles are 1.4% and 0.5%,

respectively (Fig. S9 and Table S1). Although these particles are commonly found in polluted areas from anthropogenic sources such as stationary combustion sources and vehicles (Li et al. 2016), there are almost no such local anthropogenic sources in the Arctic area. Instead, the fly-ash and Fe-aggregate particles attain better relations with the soot number fraction ($R^2$=0.35 and 0.24 for the fly-ash and Fe aggregates, respectively; Fig. S11), and soot, fly-ash, and Fe-aggregate particles are often found in the

same particles (Fig. 13 and Fig. S8). Our TEM measurements agree with the SP2 observations of Yoshida et al. (2020), who also found correlations among BC and Fe oxide during this campaign. These results indicate that the soot, fly-ash, and Fe-aggregate particles originated from anthropogenic sources through LRT.

### 3.8 Implications for particle aging and the climate

In the Arctic, although there are few anthropogenic sources and vegetation, we found that many aerosol particles originated from anthropogenic and BB sources. The soot and Fe-aggregate particles have light-absorbing properties, and the mineral-dust particles demonstrate water-insoluble, crystalline structures





with hygroscopic attachments. These optical and structural properties influence their direct radiative forcing by scattering and absorbing solar radiation and their indirect radiative forcing through cloud

modification by serving as CCN and INPs. When these light-absorbing particles are deposited on snow or ice surfaces, they reduce the surface albedo, resulting in snow/ice melting acceleration (Law and Stohl, 2007). Soot particles (or BC) have commonly been detected at various concentrations in the snow (Mori et al., 2019; Kinease et al., 2019) and ice cores (McConnell et al., 2007) in polar regions. Their deposition process from the atmosphere onto snow/ice is important to better understand their

contributions to snowmelt and albedo changes.

Soot, tarballs, and Fe-aggregate particles have been intensively investigated near their sources, such as urban areas and BB smoke. This study shows the composition and shape of aged particles (> 1 week). The soot and Fe-aggregate particles still exhibit fractal structures, especially the small particles (< several tens of primary particles; Fig. 13 and Fig. S8), similar to those found in source regions such as

East Asia (Adachi et al., 2016). This observation is inconsistent with the result showing highly compacted soot particles at a remote marine free-troposphere site (China et al., 2015). Whether soot particles become core-shell structures as they age in the atmosphere is of interest for the accurate evaluation of soot climate effects (Cappa et al., 2012; Adachi et al., 2010). Our observation implies that the soot particles in the Arctic cannot be assumed to be spherical particles but should be treated as

fractal particles when calculating the optical properties of aged samples. One explanation is that soot particles consisting of several tens or fewer primary particles hardly attain highly compact shapes because of the insufficient monomer numbers to turn them into spherical aggregates. On the other hand, most soot particles were coated with sulfate or other materials, suggesting that they possess enhanced light absorption and CCN activity. We also observed the aging of tarball particles in a BB plume, i.e.,

the surfaces of certain tarball particles had reacted with sulfate, making them hygroscopic. Aged tarballs are removed more efficiently from the atmosphere by precipitation than the original ones. As the mass fraction of tarballs in fresh BB smoke may reach ~40% (Sedlaeck et al., 2018), which is much higher than that in our samples, most tarballs could have been removed from the atmosphere during LRT. This tarball removal process has been proposed by Pósfai et al. (2004) when they first characterized tarballs,

and our study provides evidence of the above hypothesized processes.

## 4. Summary

This study reveals that the aerosol particles in the Arctic troposphere exhibit various composition, shape, and mixing state ranges depending on the sampling altitude and airmass history. Sulfate is the dominant aerosol type, and sea-salt, mineral-dust, K-bearing, and carbonaceous particles are also

observed as major aerosol species. The aerosol particles are commonly mixtures of several components, resulting in different optical and hygroscopic properties than the original particles. In addition to the





main components, they also include soot, fly-ash, and Fe-aggregate particles, all of which are solid, primary particles originating from anthropogenic sources. The aerosol particles that include soot and Fe-aggregate particles may exhibit light-absorbing properties and contribute to Arctic warming and

snow/ice melting when deposited on the surface. BB is an important source of the aerosol particles in the Arctic area. In the BB samples, we find K-bearing and tarball particles, both of which are tracers of BB emissions. Our model simulations indicated the BB contributions from Siberia. Tarballs could have aged for a week or more after their emission and have reacted with sulfate. Many mineral-dust particles are mixed with sea salt and are relatively abundant in the samples collected near the surface. The

mineral-dust occurrence contradicts previous studies that show that they mainly originate from LRT in spring, and further studies are required to confirm their sources. The ability of solid particles, such as mineral-dust, soot, fly-ash, and Fe-aggregate particles, to function as INPs should be considered to evaluate their contributions to cloud formation. This study highlights a wide range of mixing states, and the mixing states of aerosol particles after LRT should be accounted for to accurately evaluate their

climate influences.

**Data and code availability**

The TEM and simulation data used in this publication are available upon request (adachik@mri-jma.go.jp). Access to the MRI-ESM2 code is available under a collaboration framework with the MRI.

**Author contributions**

KA conducted the TEM analysis and data processing. SO, AY, and MK executed the TEM sampling and field observations. KA and NM set up the TEM sampler. MK supervised the TEM sampling. NO performed the model simulations and analyses. KA prepared the manuscript with contributions from all coauthors.

**Competing interests**

The authors declare that they have no conflicts of interest.

**Acknowledgments**

We are indebted to all the PAMARCMiP 2018 participants for their cooperation and support. The authors also acknowledge the Alfred Wegener Institute (AWI) for both the support to conduct the PAMARCMiP 2018 campaign and the use of the Polar5 research aircraft and the skill and safety

exemplified by the pilots and flight staff. We thank the financial support of the Environment Research and Technology Development Fund (JPMEERF20205001, JPMEERF20202003, JPMEERF20172003,





and JPMEERF20165005) of the Environmental Restoration and Conservation Agency of Japan, the Global Environmental Research Coordination System of the Ministry of the Environment of Japan, and the ArCS (JPMXD1300000000) and ArCS II (JPMXD1420318865) projects of the Ministry of

Education, Culture, Sports, Science, and Technology (MEXT) of Japan. KA and NO acknowledge the financial support provided by the Japan Society for the Promotion of Science (JSPS) KAKENHI (grant numbers JP26701004, JP16K16188, JP16H01772, JP18H04134, JP18H03363, JP18H05292, JP19H01972, JP19H04236, JP19K21905, and JP19H04259).





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





**Figures**

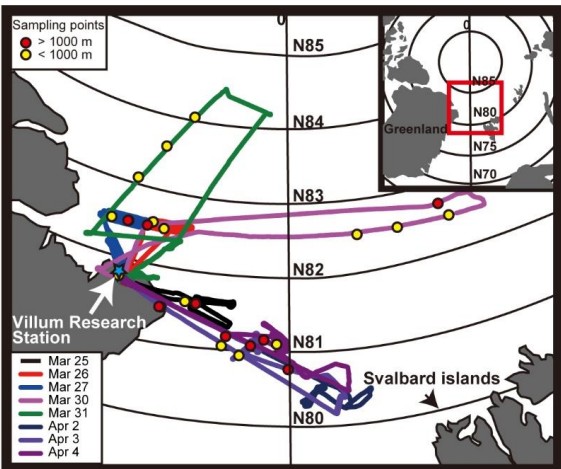

**Figure 1.** Flight tracks and locations of the TEM samplings during the PAMARCMiP 2018. The blue star indicates the Villum Research Station in Greenland. The yellow points indicate the sampling locations below 1000 m. The red points indicate those above 1000 m. The sampling points are the

midpoints of the sampling time; 19 min for all samples except during the 3 and 4 April flights, which had a sampling time of 29 min.



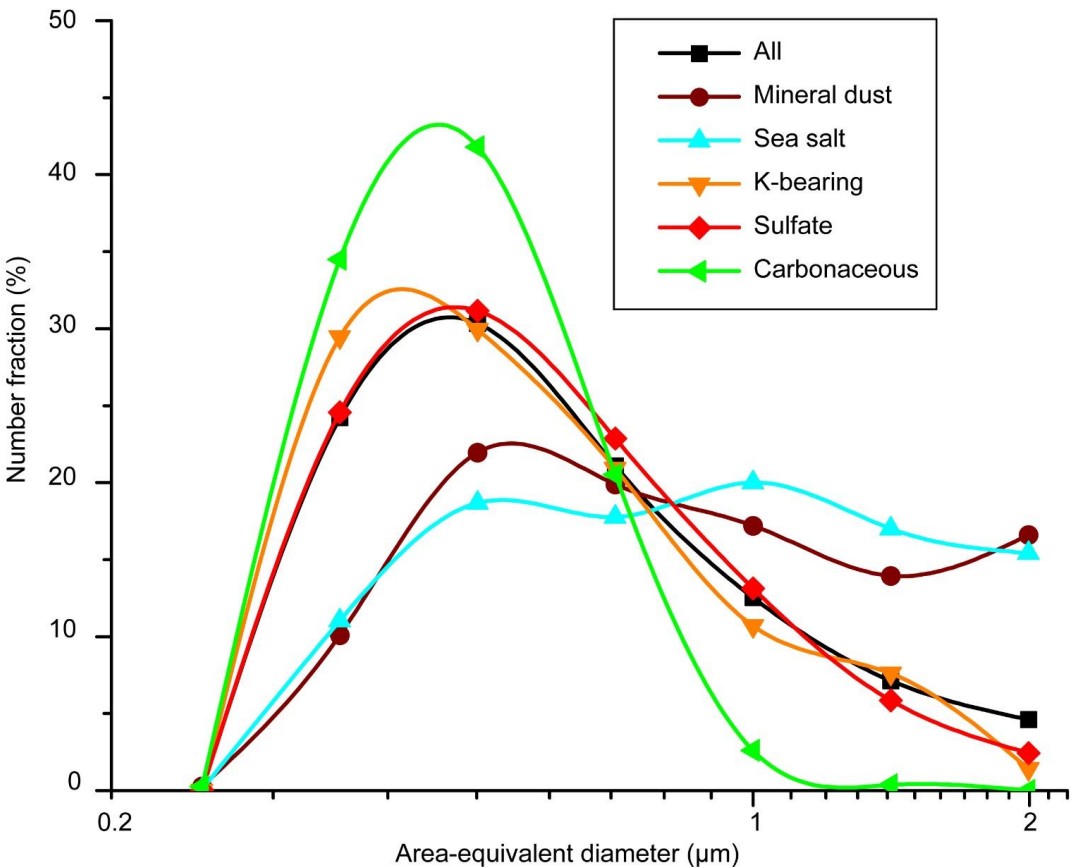

**Figure 2.** Size distributions of each particle type. The particle sizes were determined from the area-
equivalent diameters obtained from the STEM images. The size bins are shown on a log scale and are <
0.25, 0.25-0.35, 0.35-0.5, 0.5-0.71, 0.71-1.00, 1.00-1.41, and >1.41 μm. The particle numbers used for
the mineral-dust, sea-salt, K-bearing, sulfate, and carbonaceous particles are 7844, 337, 1205, 421,
4672, and 809, respectively.



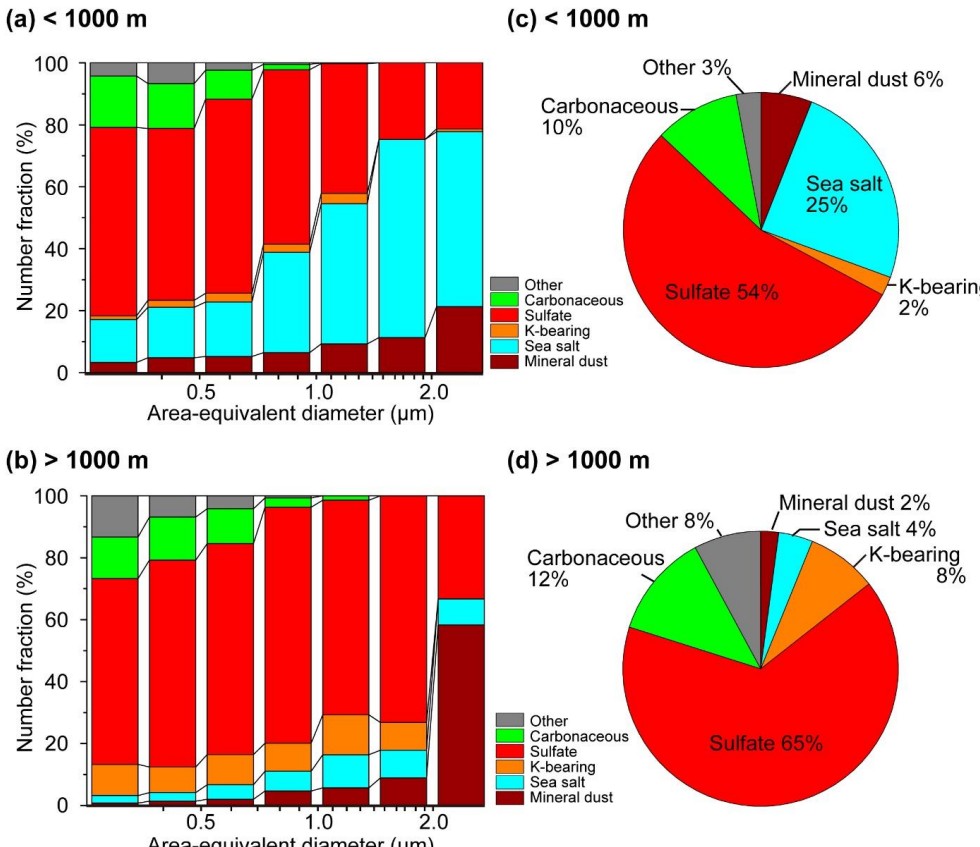

**Figure 3.** Size-dependent number fractions in the samples (a) < 1000 m and (b) > 1000 m and number fractions of the total particles of the samples (c) < 1000 m and (d) > 1000 m. The size bins in (a) and (b) are shown on a log-scale and are <0.35, 0.35-0.5, 0.5-0.71, 0.71-1.00, 1.00-1.41, 1.41-2.00, and > 2.00 μm. The particle numbers used in (c) and (d) are 4400 and 3444, respectively.





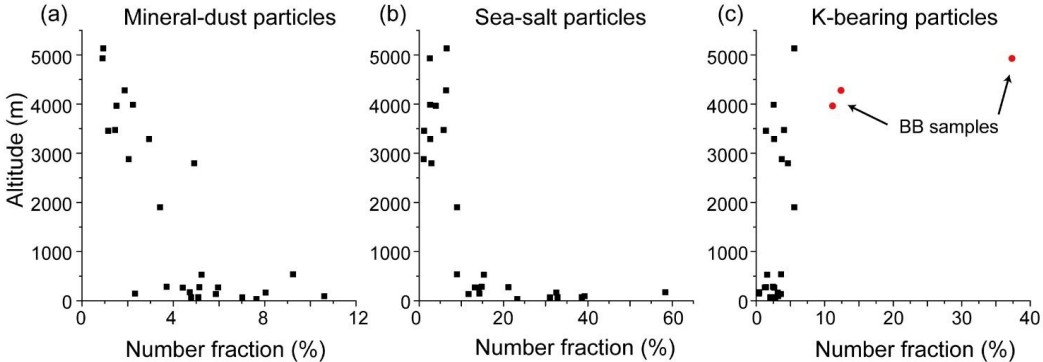


**Figure 4.** Vertical changes in the number fractions (%) of (a) mineral-dust, (b) sea-salt, and (c) K-bearing particles. Each plot indicates the number fraction of a given sample. The number of particles in the samples ranges from 216 to 442. The red solid-circle points in the K-bearing particle plots indicate the samples originating from the biomass burning plume (BB samples).






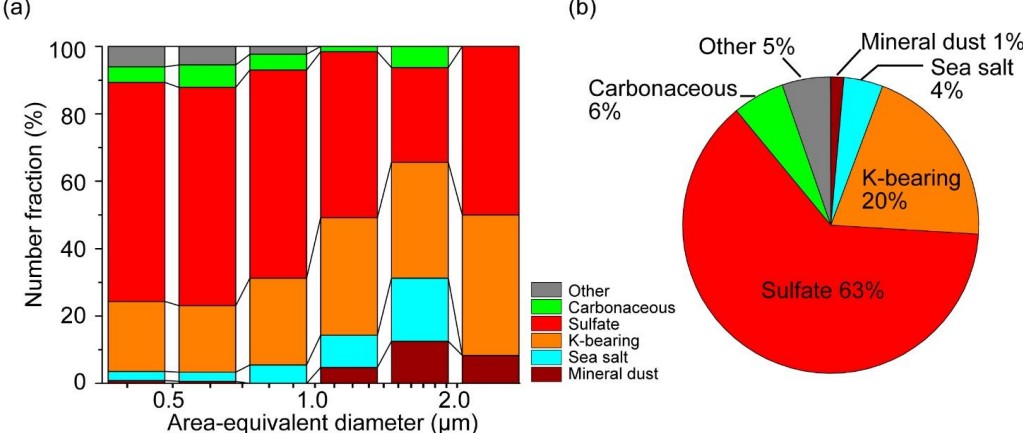

**Figure 5.** Number fractions of the aerosol particles in the BB samples. (a) Size-dependent and (b) total number fractions. Three BB samples were collected on 2, 3, and 4 April. The size bins are shown on a log-scale and are 0.35-0.5, 0.5-0.71, 0.71-1.00, 1.00-1.41, 1.41-2.00, and > 2.00 µm. (b) Number

fractions of the total particles of the BB samples. The particle number of the BB samples is 1041.




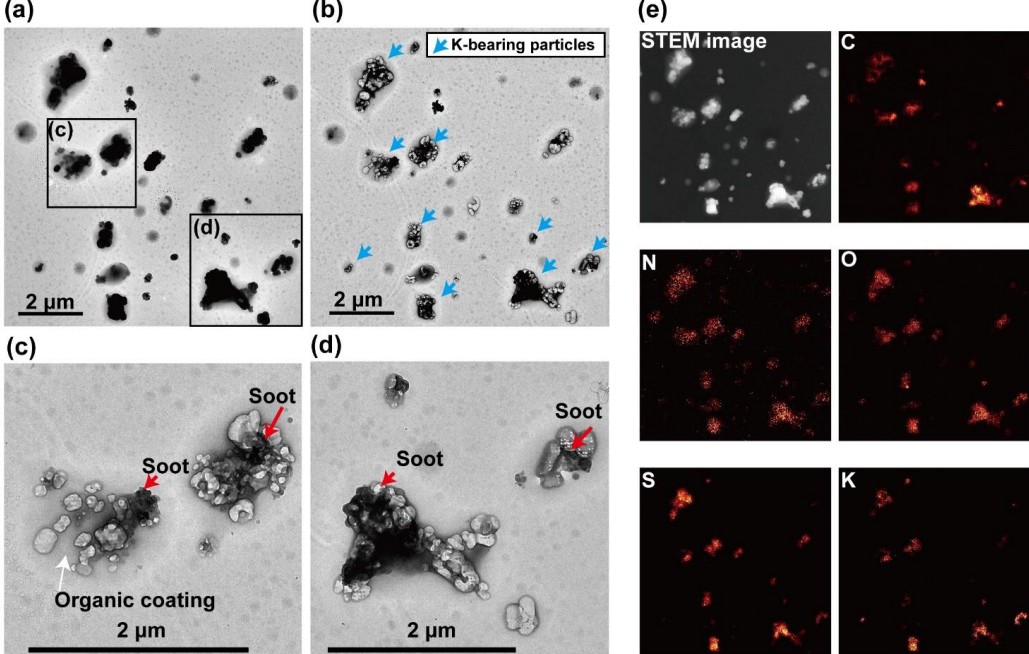

**Figure 6.** TEM and elemental mapping images of a BB sample (10:40-10:59, 2 April 2018). (a) TEM image of K-bearing particles. (b) The same area of the TEM image (a) but after the STEM-EDS

analysis. Beam-sensitive materials are removed, and soot inclusions are more apparent. (c) and (d): Examples of K-bearing particles containing soot inclusions and organic coatings, which appear in gray in the TEM image. (e) STEM and elemental mapping images for C, N, O, S, and K in the TEM image. The distributions of K, S, and O correspond to the K-bearing particles, and that of C represents the soot and organic particles.



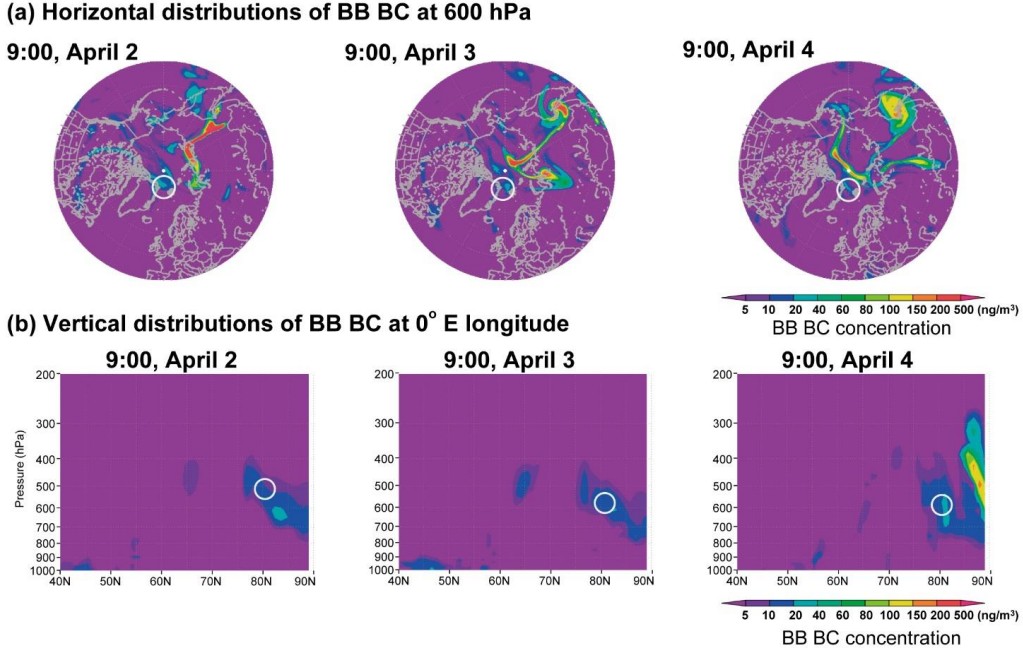


**Figure 7.** Model simulations of the (a) horizontal and (b) vertical distributions of the mass concentration of BC originating from biomass burning (BB) during the sampling periods of the BB samples. The white circles indicate the sampling locations. The transport pathway of the BB plume from its source area is shown in Fig. S2.





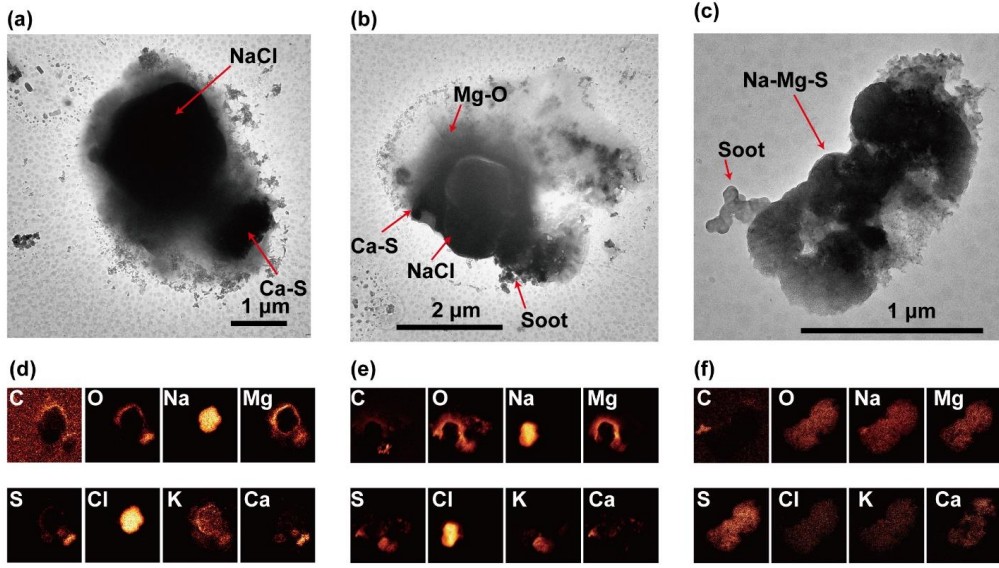


**Figure 8.** TEM and elemental mapping images of the sea-salt particles in a sample (10:00-10:19, 30 March 2018). Top images (a)-(c): TEM images of the sea-salt particles mixed with various grains. Bottom images (d)-(f): Elemental mapping images of the TEM areas for C, O, Na, Mg, S, Cl, K, and Ca. Soot particles are attached to the sea-salt particles (b) and (c). C-O-Mg in the particles (a) and (b) occurs around NaCl grains.






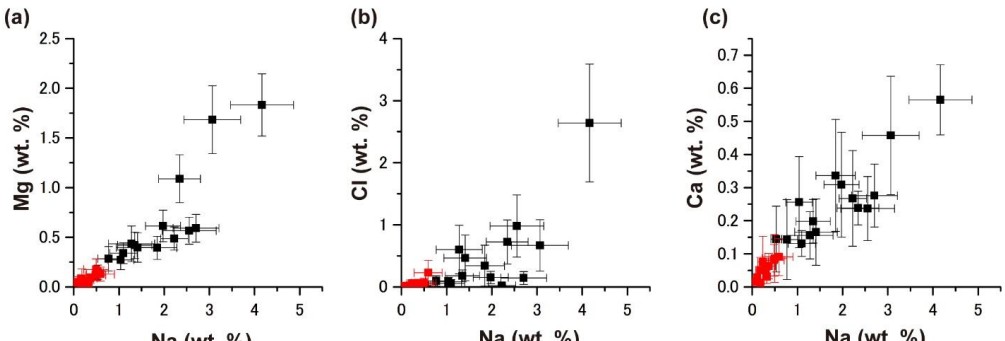

**Figure 9.** Relations of Na with Mg, Cl, and Ca among all particles. The plots show the average weight percent values of all particles within each sample. The black and red plots are the samples < 1000 m and > 1000 m, respectively. The error bars indicate the 95% confidential intervals.






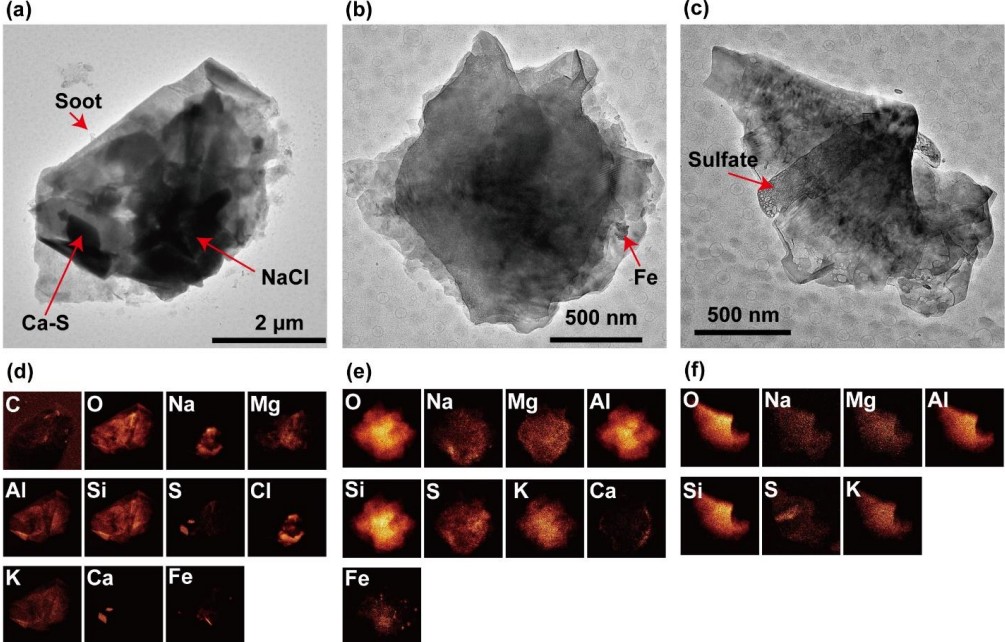

**Figure 10.** TEM and elemental mapping images of the mineral-dust particles. Top (a)-(c): TEM images
of the mineral-dust particles. Bottom (d)-(f): Elemental mapping images of the TEM areas for the
detected elements. All particles were found in the samples collected from 10:00-10:19 on 30 March
2018. Soot, NaCl, and Ca-S particles are attached to these mineral-dust particles in (a). Fe occurs as
grains in (b). Sulfate is attached to the mineral-dust particle in (c).





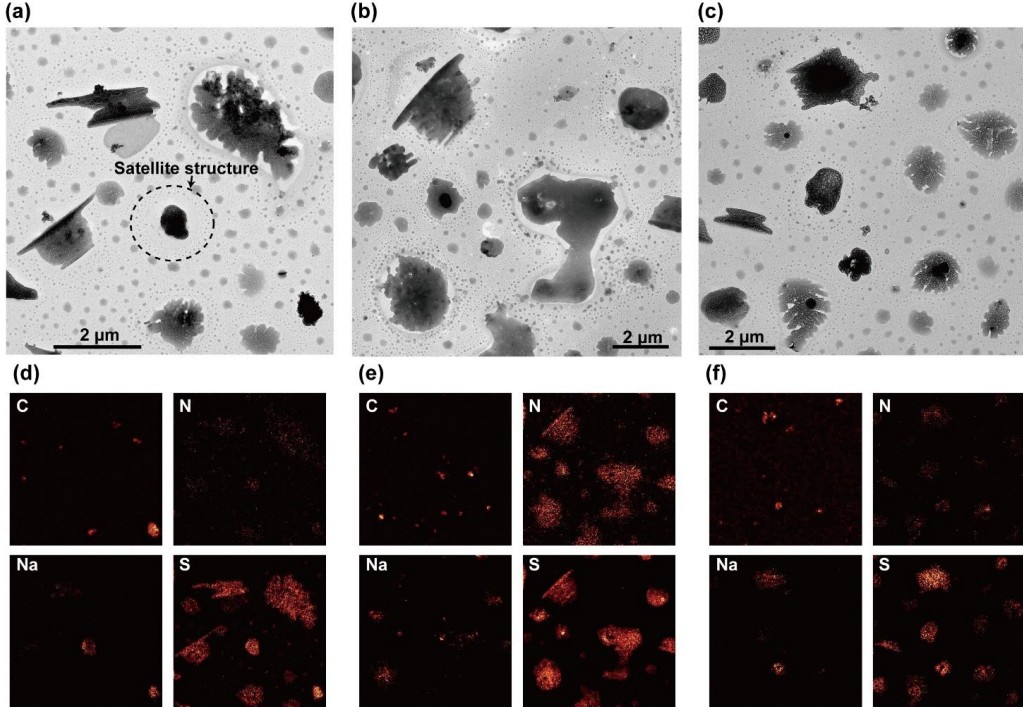

**Figure 11.** TEM and elemental mapping images of the sulfate particles. Top (a)-(c): TEM images.
Bottom (d)-(f): Elemental mapping images of the TEM areas for C, N, Na, and S. All particles were
collected from 14:40-14:59, 31 March 2018. Most particles mainly consist of sulfate. Some particles
also include N and Na, suggesting that they are mixtures of ammonium sulfate and sodium sulfate. The
carbon grains correspond to the internally mixed soot particles. Small droplets occur around relatively
large sulfate particles, forming a satellite structure, as indicated in (a).





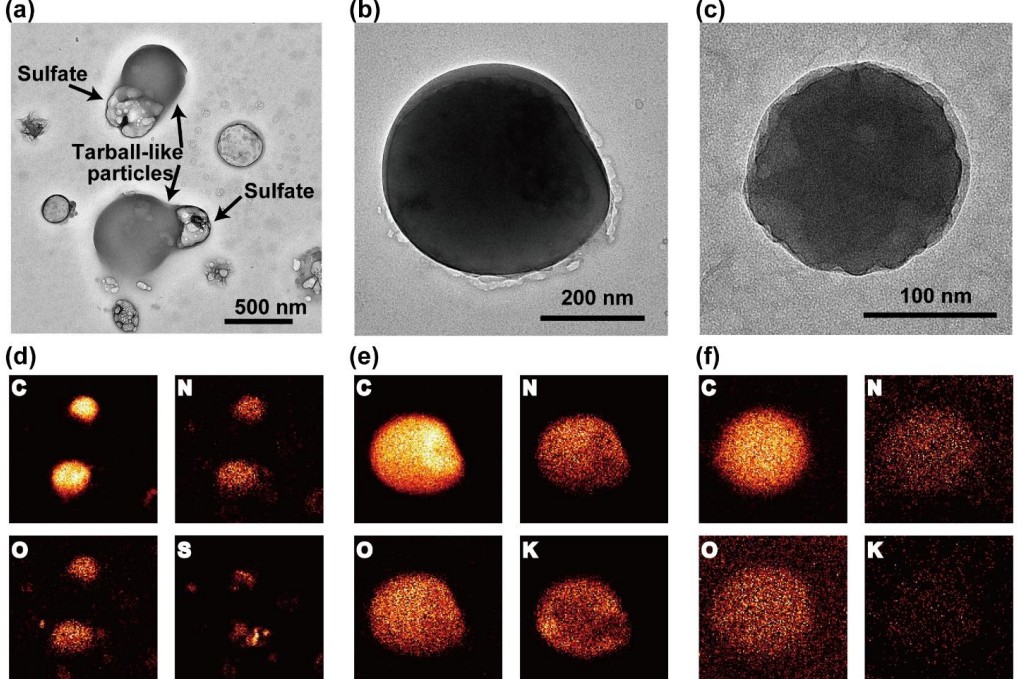

**Figure 12.** TEM and elemental mapping images of the tarball and tarball-like particles in a BB sample (10:30-10:59, 2 April 2018). (a) TEM image of the tarball-like particles containing sulfate. As they exhibit deformed spherical shapes and the characteristic elements of tarballs (e.g., C, N, O, and K), we label them tarball-like particles. Other examples of tarballs are shown in (b) and (c). (d)-(f): Elemental mapping images of the TEM areas.



**Figure 13.** TEM and elemental mapping images of the sulfate particles with inclusions. (a) TEM image before the mapping analysis and (b) after the mapping analysis. Electron-beam exposure removed sulfate, and inclusions became more apparent. (c) Elemental mapping images of the TEM areas for the detected elements. (d) A fly-ash particle. (e) Mineral-dust and fly-ash particles. (f) An Fe-aggregate particle. (g) Mineral-dust and soot particles. The white squares in (b) and (c) indicate the areas of (d)-(g). The sample was collected from 14:40-14:59, 31 March 2018.