# Peer review of "Compositions and mixing states of aerosol particles by aircraft observations in the Arctic springtime, 2018"

_Atmospheric Chemistry and Physics, 2020_

## Referee Comment (RC1) · Anonymous Referee #1 · 21 Nov 2020

Review of Adachi et al., ACP, 2020

This manuscript evaluates particle samples collected by impaction upon electron microscope grids and analyzed by transmission and scanning transmission electron microscopy with x-ray compositional analysis. The samples were collected on an aircraft in the Arctic in springtime, when the Arctic haze phenomenon is a maximum.

The methodology and analysis is very clear, and provide useful information on the mixing state, morphology, and size-dependent composition of the aerosol in the Arctic in springtime. The complexity of the aerosol is surprising, with mineral dust, sea-salt, soot, fly ash, and dust particles adding to the more widely recognized sulfate-organic

mixtures in this region. The manuscript is a carefully written and useful addition to the literature on Arctic haze, and should be suitable for publication following minor revision.

Below are major comments, followed by technical issues.

Major comments:

1) One overall disappointment is the relative lack of integration of the electron microscopy data with on-line aerosol instruments on the aircraft. For example, according to the project website, an aerosol optical particle spectrometer was operated on the aircraft during PAMARCMiP 2018, and an SP2 instrument provided information on black carbon abundance and coating thickness. Instead of combining these measurements with the TEM and STEM data, the microscopy results are analyzed and interpreted alone. Online single particle mass spectrometers, which provide statistical information on the size resolved mixing state and composition, are being combined with independently measured particle size distributions to provide a more quantitative description of the aerosol (e.g., Froyd et al., 2019; https://doi.org/10.5194/amt-12-6209-2019). For future publications (not for this one), I urge the authors to consider blending their very useful, but quantitatively limited, compositional data with online techniques to place the results on a more quantitative footing.

2) In the abstract, and elsewhere in the manuscript, the Arctic aerosol is described as being "internally mixed". However, it's clear that there are really separate aerosol types–sulfate, sea-salt, mineral-dust, K-bearing, and carbonaceous–that were in different particles. This is the definition of an external mixture. I think the authors mean to say that all of the types were coated with sulfate/organic materials, and that some particles were composed of two or more different compositions that had coagulated. But definitely one could NOT say that the aerosol was composed of a single, internally mixed composition.

3) Line 123. Is the area-equivalent diameter just the observed diameter as viewed on the microscope grids, or is there some reconstruction to a three-dimensional form from

the (flattened?) images from the microscope? Please clarify.

4) Line 139. The classifications based on fractional elemental composition seem non-specific. For example, a particle could be considered both "carbonaceous" if it has a C + O weight percent >90%, but also a "sulfate" particle if it also has a S weight percent >2%. In fact, I'd expect all carbonaceous particles to have some significant S component. Can you comment on the fraction of particles that could be ambiguously classified, and how these are resolved by your scheme?

5) Lines 168-170. How does the model represent particulate components? Is it a bulk (mass) model, or does it have a binned or modal representation of the size distribution? A little more detail (a couple of sentences) here would be very helpful.

6) Line 299. Groot Zwaaftink et al., 2016 (https://doi.org/10.1002/2016JD025482) report on local sources of dust in the Arctic, so your data are very pertinent to their hypothesis that Arctic dust emissions are substantial.

7) Lines 320-322. Is the number of "tarball" particles on each grid correlated with the number of potassium-rich particles on each grid? This would provide a useful link to biomass burning as a source of the tarballs.

8) Section 3.7. Are these particle types counted in the "other" category in e.g., Fig. 3? I'm a little confused between the detailed description of complex particle inclusions and blended types in Section 3.7 with the very discrete particle categories discussed elsewhere in the manuscript.

9) Lines 358-359, how does the soot number fraction compare with values from the SP2 instrument?

10) Please provide 2-sided linear regressions and slopes to Fig. 9. It would be better to show molar values rather than weight percents.

11) The figures are very nice and clear!

Technical comments:

a) Line 42, change to "However, air pollution over the Arctic, named 'Arctic haze', due to . . . ."

b) Line 50, change to "source regions and to altitudes above the polar dome. . ."

c) Line 64, remove hyphen between "soot" and "mixing".

d) Line 87, the acronym PAMARCMiP is already defined in the abstract.

e) Line 126-128. I don't understand this sentence regarding AED and mixing state.

f) Line 130, change "element" to "elemental"

g) References. The formatting of the references is inconsistent, and also not consistent with Copernicus guidelines. For example, journal names are not abbreviated, and some entries, such as Buseck et al., have capitalized the paper titles. Please review and manually correct; don't rely on your reference manager software.

h) Fig. 2 caption. These are normalized size distributions, not raw number concentrations.

---

## Referee Comment (RC2) · Anonymous Referee #2 · 25 Nov 2020

The study focused on the individual aerosol particles at different altitudes during the Polar Airborne Measurements and Arctic Regional Climate Model Simulation Project. Certainly, the data from this study is important and valuable due to the ARCTIC polar aerosol in particular from the aircraft data. The study found several particle types associated with continental emissions which might be interesting for potential climate and CCN study in the Arctic. For example, the authors found one BB case based on the K-bearing particles >3900 m. Also, they found several aspects on inclusions such as fly ash, soot, and Fe-aggregates particles. For my views, I would like to recommend this paper published in ACP after one minor revision.

[Figure]

L26, the ice-nucleating particles. Maybe this authors consider whether it is suitable here. I did not noticed any discussion on these. Also there particles were internally mixed with sulfate. The secondary aerosol particles become important.

L27, For the first time. As your introduction, Hara et al. had some works in Arctic air through the aircraft. Right?

L142-144, There is no need to mention the P-rich particles. In this paper, you give details of several particle types. I might suggest to delete it.

L188 About local emission, I might have confused. You need to more data to support the statements such as back trajectories. For me. It is not necessary to classify the local emission. Below 1000m, there are still large part of particles from the long range transports.

L235, as seen in the TEM results (need Figure ?)

L236-237, terms of the BB samples, the sampling areas were > 4600 km far away from the BB sources, and the samples had aged for a week or more (need Figure ?  to support)

L245-247, There are absent several important references about "organic salts" (Laskin et al., 2012;Chi et al., 2015) on aged sea salts beside nitrate and sulfate in this sentence. As you mentioned L256-257, you found C in coating. This is very significant information.

L250-251, you mentioned N in aged particles. The authors should be familiar with the N underestimated by the TEM/EDS, although there are nitrates in aged sea salts.

L280, The TEM images are not strong enough the statement about the mineral dust particles. In the Figure 10bc, Elemental Na can be existing in mineral dust particles. Also, I noticed there is no Cl in the particles. I suspected that these particles could be externally mixed particles. As you discussed below, the mineral dust particles might source from the local Arctic areas instead of the out of Arctic.

L310, interesting, the authors detected some satellite particles. Recently, one study found the satellite particles also contain organic acids (Yu et al., 2019). That mean these contain sulfuric acids and organic acids instead of only sulfuric acid. Moreover, does the study found the significant organic coating on sulfate as reported by Yu et al. on the ground in Arctic air.

L330, "suggesting that they reacted with other species and had aged during LRT.". There is no evidence to show their reaction. In previous study, there is some irregular primary organic particles.

L340-341, I might delete the sentence. Even the soot mixed with mineral dust particles . They are not inclusions.

L403, i.e., BC

L420 ALSO, the author mentioned the reactions between tarball and sulfate. How? There is no evidence. Seemly, this is very complicate question.

Figure 13, the a, b should be noted (b) after sublimed particles or beam damage? The potential readers can directly read the information a, and b.

References: Chi, J. W., Li, W. J., Zhang, D. Z., Zhang, J. C., Lin, Y. T., Shen, X. J., Sun, J. Y., Chen, J. M., Zhang, X. Y., Zhang, Y. M., and Wang, W. X.: Sea salt aerosols as a reactive surface for inorganic and organic acidic gases in the Arctic troposphere, Atmos. Chem. Phys., 15, 11341-11353, 2015.

Laskin, A., Moffet, R. C., Gilles, M. K., Fast, J. D., Zaveri, R. A., Wang, B., Nigge, P., and Shutthanandan, J.: Tropospheric chemistry of internally mixed sea salt and organic particles: Surprising reactivity of NaCl with weak organic acids, J. Geophys. Res., 117, D15302, 10.1029/2012jd017743, 2012.

Yu, H., Li, W., Zhang, Y., Tunved, P., Dall'Osto, M., Shen, X., Sun, J., Zhang, X., Zhang, J., and Shi, Z.: Organic coating on sulfate and soot particles during late summer in the Svalbard Archipelago, Atmos. Chem. Phys., 19, 10433-10446, 10.5194/acp-19-

10433-2019, 2019.

---

## Referee Comment (RC3) · Anonymous Referee #3 · 8 Dec 2020

This manuscript investigates size and composition of individual aerosol particles collected from the Arctic using transmission electron microscopy. The authors applied the Meteorological Research Institute Earth System Model to investigate the transport patterns and aerosol sources. The authors found several types of particles and compares size resolved chemical composition below and above 1000m. Overall the manuscript is well written, nice illustrative TEM images and the research topic is relevant and important for the community. However, the discussion regarding the atmospheric implications and what are the impacts of these findings need to be discussed in detail. Some of the observations are rather qualitative. The authors used the Earth System Model but it could be better utilized for discussion in relate to the particle composition and sources.

Specific comments:

Particle classification is bit confusing to me. The authors discussed about the K-bearing particles. Are those associated with carbonaceous or sulfate? Fig 6 shows some TEM images where K-bearing particles are both associated with carbonaceous and sulfate but what are their number fractions? Did you observe any size dependency of K-bearing particles, like smaller particles contain higher wt% or vice versa?

Number fraction of sulfate did not show any trend with altitude but what about their size? May be the authors can provide a plot of size of these particles as a function of altitude. Based on Figure 3 sulfate particles are smaller in size for <1000m but increases at >1000m. Please add some discussion.

As one of the focus of this manuscript is looking at size resolved particle composition at different altitude, it might be useful to add additional plot in figure 2 of size distribution of particles for <1000 m and >1000m for different particle classes.

The authors suggested that sea salt particles are mostly processed by sulfate. What about nitrate or organics? Did you observe smaller particles more processed compared to larger particles? Figure S6 can be improved by looking at the elemental ratio and size information as color code or bubble plot.

The authors describe soot, fly-ash and Fe-aggregates separately. What about their relative contributions? Soot should come under carbonaceous particle class. What are the relative number fractions of soot with respect to total carbonaceous particles? What are the number fraction of soot and tar balls in the biomass burning plumes investigated here?

I did not follow discussion about the tar balls aging. The authors discussed that "their composition is similar to that of particles from young BB smoke plumes" and later discussed that particles were probably >1 week aged and the surface of tar balls contain sulfate.

[Figure]

The authors discussed that for the same field campaign previous study found good agreement between the SP2 measurement and TEM observation. Did they observe similar number or mass fraction of Fe-bearing particles? Is the reported number fraction relative to total particles or contribution within dust category?

I suggest the authors to improve the atmospheric ageing and climatic impacts part.

---

## Author Comment (AC1) · 26 Jan 2021

Reviewers' comments are in red and bold and marked R.

Authors' comments are marked A.

*Revised and original text are in bold and italic*

Anonymous Referee #1

R1: This manuscript evaluates particle samples collected by impaction upon electron microscope grids and analyzed by transmission and scanning transmission electron microscopy with x-ray compositional analysis. The samples were collected on an aircraft in the Arctic in springtime, when the Arctic haze phenomenon is a maximum.

The methodology and analysis is very clear, and provide useful information on the mixing state, morphology, and size-dependent composition of the aerosol in the Arctic in springtime. The complexity of the aerosol is surprising, with mineral dust, sea-salt, soot, fly ash, and dust particles adding to the more widely recognized sulfate-organic mixtures in this region. The manuscript is a carefully written and useful addition to the literature on Arctic haze, and should be suitable for publication following minor revision.

**A1:** We appreciate the reviewer comments and suggestions. We also hope that our findings contribute to the understanding of the Arctic haze.

Major comments:

R1-1: One overall disappointment is the relative lack of integration of the electron microscopy data with on-line aerosol instruments on the aircraft. For example, according to the project website, an aerosol optical particle spectrometer was operated on the air- craft during PAMARCMiP 2018, and an SP2 instrument provided information on black carbon abundance and coating thickness. Instead of combining these measurements with the TEM and STEM data, the microscopy results are analyzed and interpreted alone. Online single particle mass spectrometers, which provide statistical information on the size resolved mixing state and composition, are being combined with independently measured particle size distributions to provide a more quantitative description of the aerosol (e.g., Froyd et al., 2019; https://doi.org/10.5194/amt-12-6209-2019). For future publications (not for this one), I urge the authors to consider blending their very useful, but quantitatively limited, compositional data with online techniques to place the results on a more quantitative footing.

**A1-1:** Thank you for the suggestion. We agree that the combination of the TEM off-line data and on-line data is a powerful tool for a comprehensive understanding of aerosol particles. We

add a section (3.7.3) and a figure (Fig. S12) to compare the TEM and SP2 results. We are also seeking future opportunities to combine TEM and on-line data.

*3.7.3 Comparison between TEM and SP2 data for soot/BC and Fe-bearing particles/FeOx*
*We further compared the TEM results with SP2 data reported by Yoshida et al. (2020). The TEM and SP2 measurements for soot/BC and Fe-bearing particle/FeOx reasonably correlate for non-BB samples (Fig. S12). The $R^2$ values of non-BB samples for soot/BC and Fe-bearing particle/FeOx are 0.45 and 0.33, respectively. The latter has a weaker correlation possibly because of their small number in the Arctic atmosphere. Although TEM and SP2 utilize different particle properties, i.e., composition/shapes and optical properties, respectively, their correlations assure these techniques. The BB samples, on the other hand, showed outliers from the relations, especially for the soot/BC. A possible reason is that BB samples have higher total particle number concentrations than non-BB samples, resulting in smaller number fractions in the TEM samples than SP2.*

[Figure]

*Figure S12. Relations between TEM and SP2 measurement for Soot/BC and Fe-bearing particles/FeOx. Left: Soot-bearing particle number fraction by TEM vs. BC number concentrations by SP2. Right: Fe-bearing particle number fraction vs. FeOx number concentration by SP2. Red plots indicate BB samples. The correlations were calculated for non-BB samples.*

R1-2: In the abstract, and elsewhere in the manuscript, the Arctic aerosol is described as being "internally mixed". However, it's clear that there are really separate aerosol types–sulfate, sea-salt, mineral-dust, K-bearing, and carbonaceous–that were in different particles. This is the definition of an external mixture. I think the authors mean to say that all of the types were coated with sulfate/organic materials, and that some particles were composed of two or more different compositions that had coagulated. But definitely one could NOT say that the aerosol

was composed of a single, internally mixed composition.

**A1-2:** Yes, "all of the types were coated with sulfate/organic materials, and that some particles were composed of two or more different compositions that had coagulated" is what we meant. We revised the relevant sentence in the abstract.

Original in abstract: *We found that sulfate, sea-salt, mineral-dust, K-bearing, and carbonaceous particles were the major aerosol constituents and were internally mixed.*

Revised: *We found that sulfate, sea-salt, mineral-dust, K-bearing, and carbonaceous particles were the major aerosol constituents. Many particles were composed of two or more compositions that had coagulated and were coated with sulfate, organic materials, or both.*

R1-3: Line 123. Is the area-equivalent diameter just the observed diameter as viewed on the microscope grids, or is there some reconstruction to a three-dimensional form from the (flattened?) images from the microscope? Please clarify.

**A1-3:** The former "the area-equivalent diameter just the observed diameter as viewed on the microscope grids" is correct. We added more information in the relevant sentence.

Original in section 2.2: *The particle size was measured based on the area-equivalent diameter (AED) determined from the STEM images.*

Revised: *The particle size was measured based on the area-equivalent diameter (AED), which was obtained from the particle area identified by making binary images using appropriate thresholds to distinguish them from the substrate in the STEM image.*

R1-4: Line 139. The classifications based on fractional elemental composition seem non-specific. For example, a particle could be considered both "carbonaceous" if it has a C + O weight percent >90%, but also a "sulfate" particle if it also has a S weight percent >2%. In fact, I'd expect all carbonaceous particles to have some significant S component. Can you comment on the fraction of particles that could be ambiguously classified, and how these are resolved by your scheme?

**A1-4:** We showed the particle classification flow in Figure S1. Here, a particle is classified only one of the categories. We also revised the particle classification section in the revised text to

clarify the method (section 2.3).

[Figure]

Figure S1.

Original in section 2.2: *We categorize the analyzed particles into six particle types based on their composition, i.e., mineral-dust particles (both Al and Fe > 0.2 wt.%), sea-salt particles (Na > 1 wt.%), K-bearing particles (K > 2 wt.%), sulfate particles (S > 2 wt.%), carbonaceous particles (C + O > 90 wt.%), and other particles. The workflow of particle classification is shown in Fig. S1. This classification may underestimate the particles in the lower categories in the flow chart, such as sulfate and carbonaceous particles, which partially coat or are attached to other particles.*

Revised in section 2.3.1: *We categorize the analyzed particles into six particle types based on their composition, i.e., 1. mineral-dust particles (both Al and Fe > 0.2 wt.%), 2. sea-salt particles (Na > 1 wt.%), 3. K-bearing particles (K > 2 wt.%), 4. sulfate particles (S > 2 wt.%), 5. carbonaceous particles (C + O > 90 wt.%), and 6. other particles. The workflow of particle classification is shown in Fig. S1. When a particle agrees with multiple particle type definitions, the particle is classified as the upper categories. This classification may underestimate the*

*secondary particles in the lower categories in the flow chart, such as sulfate and carbonaceous particles, which partially coat or are attached to other particles. We thus consider the occurrences of secondary particles using elemental mapping images.*

**R1-5: Lines 168-170. How does the model represent particulate components? Is it a bulk (mass) model, or does it have a binned or modal representation of the size distribution? A little more detail (a couple of sentences) here would be very helpful.**

**A1-5:** We added details of our model in section 2.3. The model is a bulk (mass) model, and the aerosols are represented by either binned or lognormal size distributions depending on particle components.

Original in section 2.3: *The model treats non-sea-salt sulfate, BC, organic carbon, sea salt, mineral dust, and aerosol precursor gases.*

Revised in 2.4: *The model calculates mass mixing ratios of non-sea-salt sulfate, BC, organic carbon, sea salt, mineral dust, and aerosol precursor gases. The size distributions of sea salt and mineral dust are divided into 10 discrete bins, while the other aerosol components are represented by lognormal size distributions.*

**R1-6: Line 299. Groot Zwaaftink et al., 2016 (https://doi.org/10.1002/2016JD025482) report on local sources of dust in the Arctic, so your data are very pertinent to their hypothesis that Arctic dust emissions are substantial.**

**A1-6:** Thank you for letting us know the interesting paper, which provides useful information to interpret our results.

Added in section 3.4: *Groot Zwaaftink et al. (2016) also showed a simulation result of substantial contributions of mineral dust from Eurasia and north America >60° N to the surface dust concentrations at the Arctic region during early spring. Their studies agree with our observations, suggesting that the mineral-dust particles originate from the Arctic sources.*

Original in summary: *The mineral-dust occurrence contradicts previous studies that show that they mainly originate from LRT in spring.*

Revised: *The mineral-dust occurrence suggests that they originated from the Arctic local*

*sources but not from LRT, and further studies are required to confirm their sources.*

**R1-7: Lines 320-322. Is the number of "tarball" particles on each grid correlated with the number of potassium-rich particles on each grid? This would provide a useful link to biomass burning as a source of the tarballs.**

**A1-7:** Tarballs were found only in the BB samples, which have high in K-bearing particles. Their occurrences imply that they correlate in our samples. However, the number of tarball particles is too small to quantitively discuss the relation (number fractions < 1% in section 3.6). To support the hypothesis of the tarball source, we added our recent paper showing the tarball transport from Siberia BB to the Arctic.

**Added in section 3.6:** *Tarballs from Siberia BB were also found over the northwestern Pacific in 2016, showing a transport of tarballs toward the Arctic region (Yoshizue et al., 2020).*

**R1-8: Section 3.7. Are these particle types counted in the "other" category in e.g., Fig. 3? I'm a little confused between the detailed description of complex particle inclusions and blended types in Section 3.7 with the very discrete particle categories discussed elsewhere in the manuscript.**

**A1-8:** Inclusions are not always categorized in the "other" categories. The host particles that embed or attach "inclusion particles" are classified into one of the aerosol categories based on their major compositions listed in Fig. S1. The inclusions are identified based on the shapes and compositions independent from the host particle classifications. We added a new section 2.3 to make this point clear.

**Added in section 2.3:** *Aerosol particles and their inclusions, if any, are classified into each particle type. First, we classified each major aerosol particle type based on its composition and measured their AED. Then, if the particles embed or attach inclusions (soot, fly-ash, and Fe-oxide aggregate), we classified them based on their compositions and shapes.*

**Revised in section 3.7:** *Independent from the classifications of their host particles, inclusions were identified based on both their composition and shape determined from the TEM images after the removal of beam-sensitive materials (e.g., sulfate) by exposure to an electron beam during the STEM-EDS analysis (Fig. 13 and Fig. S8).*

**R1-9: Lines 358-359, how does the soot number fraction compare with values from the SP2 instrument?**

**A1-9:** In the original manuscript, we did not directly compare soot number fraction in TEM with BC number concentrations in SP2. In the revised one, we compared these results in Fig S12 and section 3.7.3. Please also see R1.1 for the detail.

**R1-10: Please provide 2-sided linear regressions and slopes to Fig.    9.    It would be better to show molar values rather than weight percents.**

**A1-10:** Figure 9 was revised to show 1) atomic ratios, 2) approximated curves, and 3) $R^2$ values.

[Figure]

*Figure 9. Relations of Na with Mg, Cl, and Ca among all particles. The plots show the average atomic percent values of all particles within each sample. The black and red plots are the samples < 1000 m and > 1000 m, respectively. $R^2$ values are 0.84, 0.61, and 0.89 for (a), (b), and (c), respectively. Approximated curves are shown in the gray dashed lines. The error bars indicate the 95% confidential intervals.*

**R1-11: The figures are very nice and clear!**
**A1-11:** Thank you so much.

**Technical comments:**
**R1a: Line 42, change to "However, air pollution over the Arctic, named' Arctic haze', due to "**
**A1a:** Revised as suggested.

**R1b: Line 50, change to "source regions and to altitudes above the polar dome "**
**A1b:** Revised as follows.

*LRT particles travel along several pathways from their source regions to altitudes above the polar dome.*

**R1c: Line 64, remove hyphen between "soot" and "mixing".**

**A1c:** Revised as suggested.

**R1d: Line 87, the acronym PAMARCMiP is already defined in the abstract.**

**A1d:** We would keep the acronym explanation in the main text, as indicated by the journal instruction.

*They (abbreviation) need to be defined in the abstract and then again at the first instance in the rest of the text. (ACP instruction).*

**R1e: Line 126-128. I don't understand this sentence regarding AED and mixing state.**

**A1e:** We deleted the sentence and explained the meaning of AED in sections 2.2 and 2.3.

**Revised in section 2.2:** *The particle size was measured based on the area-equivalent diameter (AED), which was obtained from the particle area identified by making binary images using appropriate thresholds to distinguish them from the substrate in the STEM image.*

**Revised in section 2.3:** *Aerosol particles and their inclusions, if any, are classified into each particle type. First, we classified each major aerosol particle type based on its composition and measured their AED.*

**R1f: Line 130, change "element" to "elemental"**

**A1f:** Revised as suggested.

**R1g: References. The formatting of the references is inconsistent, and also not consistent with Copernicus guidelines. For example, journal names are not abbreviated, and some entries, such as Buseck et al., have capitalized the paper titles. Please review and manually correct; don't rely on your reference manager software.**

**A1g:** The reference list was manually checked and revised.

**R1h: Fig. 2 caption. These are normalized size distributions, not raw number concentrations.**

**A1h:** The figure title was revised to "*Normalized size distributions of each particle type…*",

and we added a sentence in the caption.

Added in the caption of Fig. 2: *The number fraction values indicate the percentage of particles in each bin size among each particle type.*

Anonymous Referee #2

R2: The study focused on the individual aerosol particles at different altitudes during the Polar Airborne Measurements and Arctic Regional Climate Model Simulation Project. Certainly, the data from this study is important and valuable due to the ARCTIC polar aerosol in particular from the aircraft data. The study found several particle types associated with continental emissions which might be interesting for potential climate and CCN study in the Arctic. For example, the authors found one BB case based on the K-bearing particles >3900 m. Also, they found several aspects on inclusions such as fly ash, soot, and Fe-aggregates particles. For my views, I would like to recommend this paper published in ACP after one minor revision.

**A2:** Thank you for the positive comments.

R2-1: L26, the ice-nucleating particles. Maybe this authors consider whether it is suitable here. I did not noticed any discussion on these. Also there particles were internally mixed with sulfate. The secondary aerosol particles become important.

**A2-1:** We revised the abstract to delete ice-nucleating particles and describe the suggested point.

Original in abstract: *We also provide the occurrences of solid-particle inclusions (soot, fly-ash, and Fe-aggregate particles), some of which are light-absorbing and potential ice-nucleating particles.*

Revised: *We also provide the occurrences of solid-particle inclusions (soot, fly-ash, and Fe-aggregate particles), some of which are light-absorbing particles. They were mainly emitted from anthropogenic and biomass burning sources and were embedded within other relatively large host particles.*

R2-2: L27, For the first time. As your introduction, Hara et al. had some works in Arctic air

through the aircraft. Right?

**A2-2:** "For the first time" was deleted.

R2-3: L142-144, There is no need to mention the P-rich particles. In this paper, you give details of several particle types. I might suggest to delete it.

**A2-3:** The relevant sentences were deleted from the main text. Instead, we revised the caption of Fig. S1 to add "*because of the lack of these particles.*"

R2-4: L188 About local emission, I might have confused. You need to more data to support the statements such as back trajectories. For me. It is not necessary to classify the local emission. Below 1000m, there are still large part of particles from the long range transports.

**A2-4:** We revised the sentence. "local emission" should have been "emissions from the Arctic regions." We discuss the sources of aerosol particles based on our global model in the later sections.

Original in section 3.1: *The former was mainly influenced by local emissions, and the latter included LRT aerosol particles.*

Revised: *The aerosol particles above 1000 m (above the polar dome) were more influenced by LRT than those below 1000 m.*

R2-5: L235, as seen in the TEM results (need Figure ?)

**A2-5:** We added Figs. 5 and 6 here.

Revised in section 3.2: *Although the model simulations estimated the BB contributions only for the BC concentrations, we interpret that the K-bearing particles and other BB emissions were also transported along with BC as seen in the TEM results (Figs. 5 and 6).*

R2-6: L236-237, terms of the BB samples, the sampling areas were > 4600 km far away from the BB sources, and the samples had aged for a week or more (need Figure ? to support)
A
**2-6:** We added Figs 7 and S2 here.

Revised in section 3.2: *In terms of the BB samples, the sampling points were > 4600 km away from the BB sources, and the samples had aged for a week or more (Figs. 7 and S2).*

R2-7: L245-247, There are absent several important references about "organic salts" (Laskin et al., 2012;Chi et al., 2015) on aged sea salts beside nitrate and sulfate in this sentence. As you mentioned L256-257, you found C in coating. This is very significant information.

A2-7: We added these references to discuss the organic matters in section 3.3.

Revised in section 3.3: *In the atmosphere, the composition of sea-salt particles is altered through the reactions with acidic gases, thus forming sodium sulfate or nitrate (Adachi and Buseck, 2015; Gard et al., 1998; Yoshizue et al., 2019), and with organic matters (Laskin et al., 2012; Chi et al., 2015).*

Revised in section 3.3: *Similar organic Mg-C coatings on sea-salt particles have been observed over California by Laskin et al. (2012) and in the Arctic (Chi et al., 2015).*

R2-8: L250-251, you mentioned N in aged particles. The authors should be familiar with the N underestimated by the TEM/EDS, although there are nitrates in aged sea salts.

A2-8: We discuss the possibility of the loss of nitrate in sections 3.3 as follows; *"It is also possible that the measured particle sizes are too small for nitrate to retain as a particle phase and that nitrates are lost from the TEM samples after sampling because of their high volatility."*
We also added a sentence of self-absorption of outgoing X-ray that affects the N quantification.

Added in section 2.2: *The EDS technique has a limitation for the quantifications of light elements because of their relatively high self-absorption of outgoing X-ray, and the uncertainties of the EDS measurements for C, N, O, and S were evaluated within ~5 weight % (Adachi et al., 2019).*

R2-9: L280, The TEM images are not strong enough the statement about the mineral dust particles. In the Figure 10bc, Elemental Na can be existing in mineral dust particles. Also, I noticed there is no Cl in the particles. I suspected that these particles could be externally

mixed particles. As you discussed below, the mineral dust particles might source from the local Arctic areas instead of the out of Arctic.

**A2-9:** Na can exist in both mineral dust and sea salt. The Na in Fig. 10c should be a part of the mineral as it occurs with Si and Al. On the one hand, the Na in Fig. 10a should occur as sea salt as it forms NaCl. We revised the text to clear the point.

We agree that the source of mineral dust particles originated from the Arctic region. Please also see our comments R1-6 regarding the source of mineral dust.

Original in section 3.4: *The mineral-dust particles are commonly mixed with sea-salt and sulfate particles, both of which change the mineral-dust particles from hydrophobic to hydrophilic.*

Revised in section 3.4: *The mineral-dust particles are commonly mixed with sea-salt (Fig. 10a and S6) and sulfate particles (Fig. 10c), both of which change the mineral-dust particles from hydrophobic to hydrophilic.*

Added in section 3.4: *On the one hand, long-term ground observations of the aerosol composition at station Nord have indicated small contributions from soil particles in the early spring, although they exhibit peaks in summer (Nguyen et al., 2013; Heidam et al., 1999). Groot Zwaaftink et al. (2016) also showed a simulation result of substantial contributions of mineral dust from Eurasia and north America >60° N to the surface dust concentrations at the Arctic region during early spring. Their studies agree with our observations, suggesting that the mineral-dust particles originate from the Arctic sources.*

R2-10: L310, interesting, the authors detected some satellite particles. Recently, one study found the satellite particles also contain organic acids (Yu et al., 2019). That mean these contain sulfuric acids and organic acids instead of only sulfuric acid. Moreover, does the study found the significant organic coating on sulfate as reported by Yu et al. on the ground in Arctic air.

**A2-10:** Thank you for letting us know the paper. We mentioned the possibility of organic matter as follows.

Added in section 3.5: *In addition, Yu et al. (2019) reported similar satellite structures from samples collected at an Arctic ground site (Ny-Ålesund, Svalbard Islands) during late summer*

and showed sulfate ($^{32}S^-$ and $^{16}O^-$) as well as $^{12}C^{14}N^-$ signals as a proxy of organic matter on the satellites. Although our TEM measurements did not detect C and N on the satellites, a trace amount of organic matter may be possible in the Arctic sulfates.

**R2-11: L330, "suggesting that they reacted with other species and had aged during LRT.". There is no evidence to show their reaction. In previous study, there is some irregular primary organic particles.**

**A2-11:** Although there are primary organic particles with irregular surfaces, we think our findings of tarball shapes should be mentioned here. Their source region (BB), compositions (C + N), and shapes agree with the tarball criteria (Adachi et al., 2019). Our model calculation (Figs 7 and S2) and number fractions of K-bearing particles support their BB source and the LRT history. On the other hand, we agree that we do not have direct evidence to show their reaction in the atmosphere, and we revised the sentence and avoided using "aging" for tarballs.

**Original in section 3.6:** *In contrast to the smooth spherical shapes of fresh tarballs (e.g., Li et al., 200), the surfaces of our tarballs are not smooth and contain sulfates (Fig. 12), suggesting that they reacted with other species and had aged during LRT.*

**Revised:** *In contrast to the smooth spherical shapes of fresh and several days aged tarballs (e.g., Li et al., 2003; Yoshizue et al., 2020), the surfaces of our tarballs are not smooth and contain sulfates (Fig. 12), suggesting that they reacted with sulfate during LRT.*

**R2-12: L340-341, I might delete the sentence. Even the soot mixed with mineral dust particles. They are not inclusions.**

**A2-12:** We prefer to keep this sentence because we cannot confirm that we detected all inclusion particles using this method; "*When they overlap with non-beam-sensitive materials (e.g., mineral dust and sea salt), it is difficult to detect inclusions in the TEM images. (L340-341).*"
The inclusion in this study is broadly defined as "*Inclusions are particles embedded within or attached to host particles*" (in L347). Thus, soot particles that attach to mineral dust can be inclusions in this study.

**R2-13: L403, i.e., BC**

**A2-13:** We simply use BC here as we stated "*We also use the term BC instead of soot in the model simulations and for the results from those instruments that measure BC by assuming that soot and BC are qualitatively the same material*" in section 2.3.2.

**Revised in section 3.8:** *BC has commonly been detected at various concentrations in the snow (Mori et al., 2019; Kinease et al., 2019) and ice cores (McConnell et al., 2007) in polar regions.*

**R2-14: L420 ALSO, the author mentioned the reactions between tarball and sulfate. How? There is no evidence. Seemly, this is very complicate question.**

**A2-14:** First, we shorten and moved this paragraph from 3.8 (Implication) to 3.6 (Carbonaceous particles). Second, we avoid using "aging" for tarballs but use "tarballs that have sulfate on their surface" or similar wording. Lastly, we provide evidence of the reaction between tarballs and sulfate in Fig 12 (a) and (d) by showing TEM images and elemental mappings of C and S. As the tarball implication is important to understand their fate in atmosphere, we would discuss based on our findings. Although we agree that "**this is very complicate question**" and we do not have complete evidence to fully understand the tarball aging process, we do have a piece of evidence that improves our understanding. Please also see R2-11 for our reply regarding this comment.

**Original in section 3.8:** *We also observed the aging of tarball particles in a BB plume, i.e., the surfaces of certain tarball particles had reacted with sulfate, making them hygroscopic. Aged tarballs are removed more efficiently from the atmosphere by precipitation than the original ones.*

**Revised in section 3.6:** *The tarballs having sulfate are removed more efficiently from the atmosphere by precipitation than the original ones as the sulfate absorbs water.*

**R2-15: Figure 13, the a, b should be noted (b) after sublimed particles or beam damage? The potential readers can directly read the information a, and b.**

**A2-15:** Figure 13 was revised as suggested.

Anonymous Referee #3

R3: This manuscript investigates size and composition of individual aerosol particles collected from the Arctic using transmission electron microscopy. The authors applied the Meteorological Research Institute Earth System Model to investigate the transport pat- terns and aerosol sources. The authors found several types of particles and compares size resolved chemical composition below and above 1000m. Overall the manuscript is well written, nice illustrative TEM images and the research topic is relevant and important for the community. However, the discussion regarding the atmospheric implications and what are the impacts of these findings need to be discussed in detail. Some of the observations are rather qualitative. The authors used the Earth System Model but it could be better utilized for discussion in relate to the particle composition and sources.

**A3:** Thank you for the useful comments to improve our manuscript. We revised the entire discussion in each section in addition to section 3.8 (implication). We revised figures to show more quantitative information (e.g., Fig. 9 and S12). However, although we try to show our result as quantitative as possible, particle shapes and mixing states are difficult to show quantitatively. Thus, we show as many TEM images as possible to provide an idea of how these particles look like. Although imaging technique is qualitative, particle images and mixing states will provide unique information that improves our understanding of aerosol particles.
Our model results provide valuable information regarding their sources and transport pathways, especially the BB events. The model results are shown in Figs. 7, S2, S5, and S7 to evaluate the particle sources. Our model calculates mass mixing ratios of each component and does not simulate individual particle compositions, making a direct comparison with TEM analyses difficult. Thus, this study uses the model only for the evaluation of the source and transport pathways. We added details of the model description in section 3.4.
Overall, we believe that our manuscript was improved by these general suggestions. Please see the changes in our replies to the specific comments.

Specific comments:
R3-1: Particle classification is bit confusing to me. The authors discussed about the K- bearing particles. Are those associated with carbonaceous or sulfate? Fig 6 shows some TEM images where K-bearing particles are both associated with carbonaceous and sulfate but what are their number fractions? Did you observe any size dependency of K-bearing particles, like smaller particles contain higher wt% or vice versa?

**A3-1:** Detailed description of particle classification was revised in new section 2.3. Please also see our reply A1-4. K-bearing particles mostly form potassium sulfate with organic coatings (Fig. 6). They cannot be simply distinguished into potassium sulfate and potassium-bearing organic because the ratio between S and C within K-bearing particles changed continuously (Fig. 3-1-1). A figure showing the size-dependency of K-bearing particles indicates that K wt. % seems to increase when the particle sizes decrease (Fig. 3-1-2). We would show these figures only in this open discussion and not show them in the main text to focus on the current discussion.

**Added in section 2.3:**

*2.3 Particle classifications*

*Aerosol particles and their inclusions, if any, are classified into each particle type. First, we classified each major aerosol particle type based on its composition and measured their AED. Then, if the particles embed or attach inclusions (soot, fly-ash, and Fe-oxide aggregate), we classified them based on their compositions and shapes.*

**Added in section 2.3.1:** *When a particle agrees with multiple particle type definitions, the particle is classified as the upper categories. This classification may underestimate the secondary particles in the lower categories in the flow chart, such as sulfate and carbonaceous particles, which partially coat or are attached to other particles. We thus consider the occurrences of secondary particles using elemental mapping images.*

[Figure]

Figure 3-1-1. Ratios of S/C in all K-bearing particles.

[Figure]

Figure 3-1-2. Size-dependent K weight % within K-bearing particles.

R3-2: Number fraction of sulfate did not show any trend with altitude but what about their size? May be the authors can provide a plot of size of these particles as a function of altitude. Based on Figure 3 sulfate particles are smaller in size for <1000m but increases at >1000m. Please add some discussion.
As one of the focus of this manuscript is looking at size resolved particle composition at different altitude, it might be useful to add additional plot in figure 2 of size distribution of particles for <1000 m and >1000m for different particle classes.

A3-2: All particle types except mineral dust particles have smaller particle size distributions in samples from above 1000 m than those below 1000 m. We showed the altitude-dependency on their size distributions in the revised Fig. 2 and added discussion in section 3.1.

[Figure]

*Figure 2. Normalized size distributions of each particle type for all samples (left) and for samples at altitudes above and below 1000 m (right). The particle sizes were determined from the area-equivalent diameters obtained from the STEM images. The size bins are shown on a log scale and are < 0.25, 0.25-0.35, 0.35-0.5, 0.5-0.71, 0.71-1.00, 1.00-1.41, and >1.41 μm. The number fraction values indicate the percentage of particles in each bin size among each particle type. The particle numbers used for the mineral-dust, sea-salt, K-bearing, sulfate, and carbonaceous particles are 7844, 337, 1205, 421, 4672, and 809, respectively.*

Added in section 3.1: *Samples above 1000 m had smaller size distributions for all particle types except mineral dust particles than those below 1000 m (Fig. 2). This result is consistent with that by Yamanouchi et al. (2005), who showed decreases of larger particles (200-1000 nm) at a higher altitude, whereas small particles (100-200 nm) did not largely show an altitude dependency. An explanation is that large particles are neither transported for long distances nor uplifted toward high altitudes.*

**R3-3: The authors suggested that sea salt particles are mostly processed by sulfate. What about nitrate or organics? Did you observe smaller particles more processed compared to larger particles? Figure S6 can be improved by looking at the elemental ratio and size information as color code or bubble plot.**

**A3-3:** From our observation, we conclude that sea salt particles are mostly processed by sulfate. The relation between Na and S*2 + Cl in sea salt particles show a positive relationship with excess S (Fig. 3-3-1). The excess S could originate from sulfate coagulations and contributions from other cations (e.g., Mg and Ca). If nitrate largely contributed to sea salt modification,

the relation would have shown Na rich. The addition of N in the X-axis does not largely change the plot because of smaller averaged atomic ratios of N (0.6 in atomic %) than sulfur (6.2 in atomic %). We also did not detect N in the mapping data. These results indicate that sea salt particles are mostly processed by sulfate. Although it is still possible that nitrate, especially large nitrate, reacts with sea salt, we do not have evidence of the reaction in this study. We discussed the possibility of nitrate in section 3.3 as follows.

" *Although nitrate can react with sea-salt particles, N was rarely detected in our sea-salt particles, possibly because the nitrate fraction relative to that of sulfate is limited in spring (Brock et al., 2011; Fenger et al., 2013). It is also possible that the measured particle sizes are too small for nitrate to retain as a particle phase and that nitrates are lost from the TEM samples after sampling because of their high volatility.* "

Organic materials can occur with sea salt. We discussed the possibility in section 3.3. Please also see our reply A2-7.

**Revised in section 3.3:** *In the atmosphere, the composition of sea-salt particles is altered through the reactions with acidic gases, thus forming sodium sulfate or nitrate (Adachi and Buseck, 2015; Gard et al., 1998; Yoshizue et al., 2019), and with organic matters (Laskin et al., 2012; Chi et al., 2015).*

We revised Fig. S6 with size information using color code (Fig. 3-3-2). However, we did not find useful information on the sizes, possibly because the sizes are mainly determined by the hosted mineral dust particles. We revised Fig. S6 using atomic %.

Lastly, we consider the effects of sea-salt size on their modification and do not find a size effect (Figure 3-3-3). A possible reason is that the sea salt modification was largely influenced by the sampling altitude rather than the particle sizes in our samples (Fig. S4).

[Figure]

Figure 3-3-1. AA scatter plot between Na and S*2+Cl in sea salt particles. The red line indicates 1:1 line.

[Figure]

Figure 3-3-2. Atomic percent of Cl and Na within all mineral-dust particles along with their sampling altitude with particle sizes. The color code indicates particle sizes.

[Figure]

**Figure S6.**

[Figure]

Figure 3-3-3. A scatter plot between S/(S+Cl) and size in sea salt particles.

R3-4: The authors describe soot, fly-ash and Fe-aggregates separately. What about their relative contributions? Soot should come under carbonaceous particle class. What are the

relative number fractions of soot with respect to total carbonaceous particles? What are the number fraction of soot and tar balls in the biomass burning plumes investigated here?

**A3-4:** The inclusions mostly occupy small fractions of their host particles (<10% in volume; Fig. 13), and they hardly affect the classification of major aerosol types (section 2.3.1). For example, a sulfate particle that includes soot will be classified as sulfate category, followed by the criteria in Fig. S1. Only when a soot particle has no coating, it is classified as a carbonaceous type (13 % of carbonaceous particles consist of soot with thin or no coatings). Please note that major aerosol types are classified based only on their compositions (Fig. S1), and the inclusions are classified based on both shapes and compositions, i.e., soot and organic particles are both carbonaceous, but their shapes and structures are clearly different. We clarified our particle classification method in the revised text (new section 2.3).

The particle number fraction that includes soot is 17%, and the carbonaceous particle fraction is 11% (Table S1). Here, many soot particles occur in sulfate particles, and 13% of carbonaceous particles consist mainly of soot particles (section 3.6). The number fractions that include soot are 14, 12, and 14 % for April 2, 3, and 4 BB samples (Table S1). Tarball number factions are < 1% in all particles, and many are modified (Fig. 12). Thus, we hesitate to discuss their number fraction in detail but show their detailed images in Fig. 12.

**Added in section 2.3:** *Aerosol particles and their inclusions, if any, are classified into each particle type. First, we classified each major aerosol particle type based on its composition and measured their AED. Then, if the particles embed or attach inclusions (soot, fly-ash, and Fe-oxide aggregate), we classified them based on their compositions and shapes.*

**R3-5:** I did not follow discussion about the tar balls aging. The authors discussed that "their composition is similar to that of particles from young BB smoke plumes" and later discussed that particles were probably >1 week aged and the surface of tar balls contain sulfate.

**A3-5:** The text was revised. We meant that these tarballs have their characteristic compositions and sulfate coating.

**Original in section 3.6:** *These tarballs mainly consist of C and O and include some N and K (Fig. 12), and their composition is similar to that of particles from young BB smoke plumes (e.g., Pósfai et al., 2004; Adachi and Buseck, 2011).*

**Revised:** *These tarballs mainly consist of C and O and include some N and K (Fig. 12). Their*

*main composition is similar to that of particles from young BB smoke plumes (e.g., Pósfai et al., 2004; Adachi and Buseck, 2011), but, different from the young ones, some had sulfate coatings (Fig. 12).*

**R3-6: The authors discussed that for the same field campaign previous study found good agreement between the SP2 measurement and TEM observation. Did they observe similar number or mass fraction of Fe-bearing particles? Is the reported number fraction relative to total particles or contribution within dust category?**

**A3-6:** We added section 3.7.3 to discuss the comparison between TEM and SP2 results. Qualitatively, they reasonably agreed. However, because TEM and SP2 measured the number fractions and number concentrations, respectively, a quantitative comparison was difficult. We discussed them in section 3.7.3. The number fractions of Fe-aggregate inclusions are relative to total particles. We added a sentence in the caption.

Added in section 3.7.3:
*3.7.3 Comparison between TEM and SP2 data for soot/BC and Fe-bearing particles/FeOx*
*We further compared the TEM results with SP2 data reported by Yoshida et al. (2020). The TEM and SP2 measurements for soot/BC and Fe-bearing particle/FeOx reasonably correlate for non-BB samples (Fig. S12). The $R^2$ values of non-BB samples for soot/BC and Fe-bearing particle/FeOx are 0.45 and 0.33, respectively. The latter has a weaker correlation possibly because of their small number in the Arctic atmosphere. Although TEM and SP2 utilize different particle properties, i.e., composition/shapes and optical properties, respectively, their correlations assure these techniques. The BB samples, on the other hand, showed outliers from the relations, especially for the soot/BC. A possible reason is that BB samples have higher total particle number concentrations than non-BB samples, resulting in smaller number fractions in the TEM samples than SP2.*

Added in caption Table S1: *Inclusion % values are relative to total measured particles.*

**R3-7: I suggest the authors to improve the atmospheric ageing and climatic impacts part.**

**A3-7:** We revised the section. Discussion about tarball was moved to section 3.6.

Added: *As most soot particles are embedded with or attached to other species, mostly hygroscopic ones, they will be more easily removed from the atmosphere than those without*

*coatings.*

*Overall effects of soot shapes and coatings observed in this study are mixtures of both positive and negative on its radiative forcing evaluations.*